# Reciprocal priming between receptor tyrosine kinases at recycling endosomes orchestrates cellular signalling outputs

Michael P Smith[1,†], Harriet R Ferguson[1,†] (ID), Jennifer Ferguson[1], Egor Zindy[2,‡], Katarzyna M Kowalczyk[1,§], Thomas Kedward[3], Christian Bates[1], Joseph Parsons[3], Joanne Watson[4], Sarah Chandler[1] (ID), Paul Fullwood[1], Stacey Warwood[5], David Knight[5], Robert B Clarke[3,6] & Chiara Francavilla[1,6,*] (ID)

## Abstract

Integration of signalling downstream of individual receptor tyrosine kinases (RTKs) is crucial to fine-tune cellular homeostasis during development and in pathological conditions, including breast cancer. However, how signalling integration is regulated and whether the endocytic fate of single receptors controls such signalling integration remains poorly elucidated. Combining quantitative phosphoproteomics and targeted assays, we generated a detailed picture of recycling-dependent fibroblast growth factor (FGF) signalling in breast cancer cells, with a focus on distinct FGF receptors (FGFRs). We discovered reciprocal priming between FGFRs and epidermal growth factor (EGF) receptor (EGFR) that is coordinated at recycling endosomes. FGFR recycling ligands induce EGFR phosphorylation on threonine 693. This phosphorylation event alters both FGFR and EGFR trafficking and primes FGFR-mediated proliferation but not cell invasion. In turn, FGFR signalling primes EGF-mediated outputs via EGFR threonine 693 phosphorylation. This reciprocal priming between distinct families of RTKs from recycling endosomes exemplifies a novel signalling integration hub where recycling endosomes orchestrate cellular behaviour. Therefore, targeting reciprocal priming over individual receptors may improve personalized therapies in breast and other cancers.

**Keywords** fibroblast growth factor receptor; quantitative phosphoproteomics; receptor tyrosine kinases; signalling; trafficking
**Subject Categories** Membranes & Trafficking; Proteomics; Signal Transduction

The EMBO Journal (2021) 40: e107182
See also: **BP Ceresa** (July 2021)

## Introduction

Receptor tyrosine kinases (RTKs), such as fibroblast growth factor (FGF) and epidermal growth factor (EGF) receptors, respond to perturbations in the environment by initiating signalling cascades in proximity to the plasma membrane upon binding of their growth factors (Wintheiser & Silberstein, 2020). Ligand-induced proximal or early signalling is then amplified through cascades—such as the mitogen-activated protein kinase (ERK, p38), phosphoinositide 3-kinase (PI3K) and phospholipase Cγ (PLCγ) pathways—specifies cell fate and controls cellular homeostasis during development and in pathological conditions (Lemmon & Schlessinger, 2010). Indeed, deregulated early signalling and signalling rewiring are responsible for unwanted long-term outputs, such as increased proliferation and motility, in human diseases, including cancer (Du & Lovly, 2018). This is the main reason why signalling molecules are the target of most of the known cancer therapies (Yaffe, 2019). However, each RTK is not an isolated entity on the plasma membrane, but functions within complex networks with other RTKs to fine-tune cancer cell intracellular signalling and long-term fate decisions. Indeed, targeting single molecules is often not enough to switch off unwanted cancer cell responses, as highlighted for instance in breast cancer (Harbeck *et al*, 2019). Recent advances in mass spectrometry (MS)-based phosphoproteomics allows us to simultaneously analyse thousands of signalling molecules and their post-translational modifications (PTMs) (Huang, 2012; Doll *et al*, 2019; Lundby *et al*, 2019; Bludau & Aebersold, 2020). However, how different RTK families

---

1  Division of Molecular and Cellular Function, School of Biological Science, Faculty of Biology Medicine and Health (FBMH), The University of Manchester, Manchester, UK
2  Division of Cell Matrix and Regenerative Medicine, School of Biological Science, FBMH, The University of Manchester, Manchester, UK
3  Division of Cancer Sciences, School of Medical Science, FBMH, The University of Manchester, Manchester, UK
4  Division of Evolution and Genomic Sciences, School of Biological Science, FBMH, The University of Manchester, Manchester, UK
5  Bio-MS Core Research Facility, FBMH, The University of Manchester, Manchester, UK
6  Manchester Breast Centre, Manchester Cancer Research Centre, Manchester, UK
   *Corresponding author. Tel: +44 161 275 5208; E-mail: chiara.francavilla@manchester.ac.uk
   †These authors contributed equally to this work
   ‡Present address: Center for Microscopy and Molecular Imaging, Université Libre de Bruxelles (ULB), Gosselies, Belgium
   §Present address: Department of Biochemistry, University of Oxford, Oxford, UK

integrate their downstream signalling has not been comprehensively analysed yet. A better understanding of the molecular mechanisms used by RTKs to coordinate each other's signalling architecture will help to identify how to interfere with the right target at the right time to re-direct deregulated cancer cell behaviours. This idea would also support the current therapeutic concept in breast cancer aiming at personalized combination of therapies at early stage of treatment prior to the acquisition of resistance or to delay it (Harbeck *et al*, 2019).

Multiple mechanisms contribute to shape signalling architecture, including the nature and affinity of the receptor ligand (Zinkle & Mohammadi, 2019), receptor co-activation (Tan *et al*, 2017), feedback mechanisms (Nguyen & Kholodenko, 2016) and the spatiotemporal distribution of signalling transducers (Bergeron *et al*, 2016). Ligand-dependent endocytic trafficking (hereafter trafficking) of RTKs from and to the plasma membrane regulates early signalling and downstream responses (Goh & Sorkin, 2013; Schmid, 2017; Budick-Harmelin & Miaczynska, 2018). For instance, we have shown that fibroblast growth factor 10 (FGF10) and transforming growth factor α (TGFα) initiate specific early signalling events that regulate FGFR2b and EGFR recycling to the plasma membrane, respectively, resulting in enhanced cell motility (Francavilla *et al*, 2013; Francavilla *et al*, 2016). It is, however, unknown whether ligand-induced recycling plays a role in the signalling interplay between RTKs to coordinate long-term responses. This concept is supported by alterations in the ability of cancer cells to migrate and proliferate due to derailed RTK trafficking (Mellman & Yarden, 2013; Lanzetti & Di Fiore, 2017) and by EGFR/integrin recycling-dependent regulation of cancer cell migration (Caswell & Norman, 2008).

Here, to study how ligand-induced trafficking—and more specifically recycling to the plasma membrane—affected signalling coordination downstream from FGFRs in breast cancer cells we combined quantitative phosphoproteomics and targeted assays. FGFRs are a large family of RTKs composed of alternatively spliced isoforms of four genes (*Fgfr1b-c*, *Fgfr2b-c*, *Fgfr3b-c*, *Fgfr4*), where the "b" and "c" isoforms are expressed mainly on epithelial and mesenchymal cells, respectively (Ornitz & Itoh, 2015). More than 22 FGF ligands exist (Ornitz & Itoh, 2015), making FGF/FGFR an ideal system to study trafficking-dependent signalling integration. FGFRs play an important yet understudied role in breast cancer and are deregulated in a significant percentage of the oestrogen/progesterone receptor (ER/PR) and triple-negative breast cancer (TNBC) subtypes of breast cancer (Babina & Turner, 2017; Navid *et al*, 2020). Furthermore, FGFs are essential for proficient breast cancer organoid growth (Sachs *et al*, 2018), but their role *in vivo* is less clear (Clayton & Grose, 2018; Watson & Francavilla, 2018; Navid *et al*, 2020). Although breast cancer is known to have deregulated RTK signalling and trafficking (Butti *et al*, 2018), the functional consequences of recycling in breast cancer cells are yet to be determined. It is also unclear how ligand, receptor, and protein adaptors integrate trafficking and signalling to fine-tune breast cancer cellular responses. Therefore, to dissect recycling-dependent integration of signalling outputs from multiple angles, we developed three quantitative trafficking phosphoproteomics approaches (TPAs), which focused on recycling FGFs-, recycling FGFRs- and recycling adaptor-dependent signalling (Fig 1A). To our knowledge, previous global studies of the trafficking-signalling enigma focused on a single question, either uncovering general trafficking regulators by genetic screening or

multiparametric imaging analysis (Collinet *et al*, 2010; Liberali *et al*, 2014; Gut *et al*, 2018) or revealing the partners of trafficking effectors by proximity labelling methods (Gillingham *et al*, 2019). Here, we constructed for the first time a snapshot of FGFR recycling-dependent signalling in breast cancer cells and provided a comprehensive resource for the trafficking, signalling and cancer communities. We unexpectedly identified a novel signalling interplay between FGFR and EGFR from the recycling endosomes. Specifically, FGFR recycling ligands induce phosphorylation on EGFR at the non-catalytic threonine T693 (T669 in the UniProt sequence P00533) that reciprocally affects both FGFR and EGFR signalling outputs in breast cancer cells. We have therefore elucidated reciprocal priming between FGFR and EGFR which is based on an early phosphorylation-dependent signal and which coordinates trafficking-dependent signalling outputs.

# Results

## TPAs unmask FGFR-dependent EGFR_T693 phosphorylation

To investigate recycling-dependent signalling in breast cancer cells, we compared the FGFR2b recycling stimulus FGF10 to FGF7, which induces FGFR2b degradation (Francavilla *et al*, 2013) (Fig 1B and C, Appendix Fig S1A and B), and analysed changes in the global signalling of a panel of five FGFR2b-expressing breast cancer cell lines (Francavilla *et al*, 2013) (Appendix Fig S1C). The breast cancer cell lines were treated with FGF7 or FGF10 for 1 or 8 min., and such early signalling was analysed using phosphoproteomics, hereafter referred to as trafficking phosphoproteomics approach 1 (TPA1) (Fig 1A). TPA1 showed a high degree of reproducibility in four independent experiments with the identification of phosphorylated peptides in the high intensity range and the quantification of 4559 proteins and 9494 phosphorylated sites in total, consistent with previous publications (Lundby *et al*, 2019) (Appendix Fig S1C–I, Datasets EV1 and EV2). Hierarchical clustering of the differentially regulated phosphorylated sites showed clustering of breast cancer cell lines based on their known molecular subtypes (Neve *et al*, 2006), within which we identified clusters based on FGFR2b-specific early signalling (Appendix Fig S2A). We focused on the 5 cell line-specific FGFR2b signalling clusters identified across the breast cancer cell lines (32.5% of the phosphoproteome) and found an enrichment for proteins involved in signalling pathways, adhesion and establishment of localization (endocytosis and transport) common to all cell lines (Appendix Fig S2B). Within these 5 clusters, 78 kinases were identified as being phosphorylated (Appendix Fig S2C) and 12 of them were associated with the Gene Ontology (GO) terms establishment of localization and/or adhesion —including EGFR—consistent with enrichment of all proteins (Appendix Fig S2B and C). To study the differential phosphoproteomes downstream from FGF7 and FGF10, we focused on T47D and BT20. Each cell line represents a distinct breast cancer molecular subtype (Neve *et al*, 2006), mirrored in FGFR2b-specific clusters (Appendix Fig S2A). Furthermore, these cell lines express different levels of *Fgfr2b* (Appendix Fig S2D) and respond to FGF7/10 stimulation, as shown by increasing phosphorylation of ERK and cell proliferation (Watson & Francavilla, 2018) (Fig 1D, Appendix Fig S2E–G). Interestingly, hierarchical clustering showed that each FGF

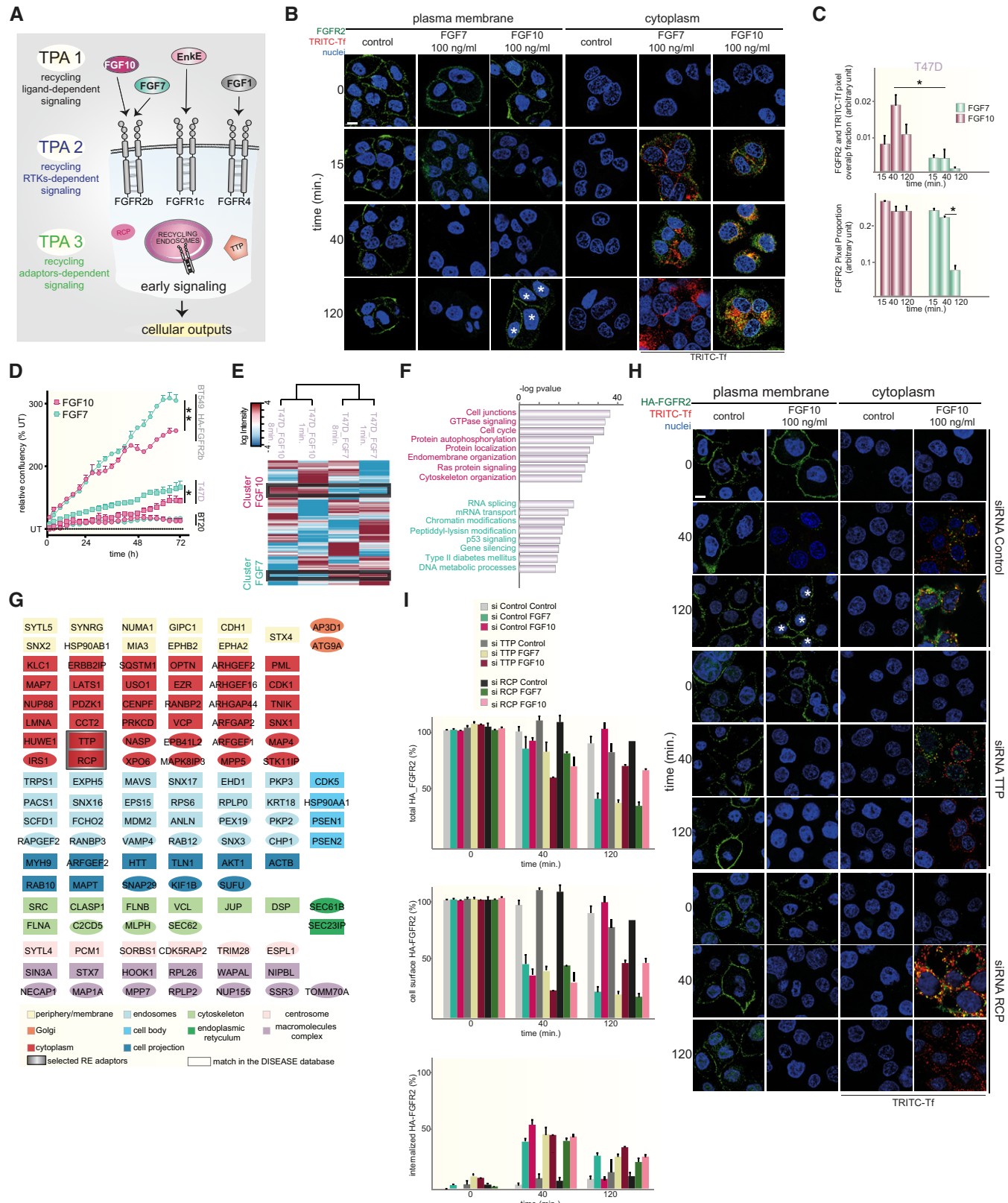

**Figure 1.**

◄

**Figure 1. Trafficking phosphoproteomics reveals FGFR2b recycling-dependent outputs.**

A  Overview of the trafficking phosphoproteomic approaches (TPAs).
B  Internalization (cytoplasm) and recycling (plasma membrane) of FGF7- and FGF10-stimulated endogenous FGFR2 (green) for 0, 15, 40, and 120 min. in T47D. TRITC-Tf is a marker of recycling (red). Nuclei are stained in blue. *, cells with receptor recycled to the plasma membrane. Scale bar, 5 μm.
C  Red and green pixels overlap fraction representing the co-localization of FGFR2b with the recycling endosomes marker Tf (above) and the proportion of green over total pixels representing FGFR2b in the cytoplasm (below) upon stimulation with FGF7 (dark green) or FGF10 (burgundy) for 15, 40, or 120 min. Values represent the median ± SD of at least 3 independent experiments. Representative pictures are shown in B (cytoplasm). *P < 0.005 (Student's t-test).
D  Confluence over time of BT20, T47D, and BT549 transfected with HA-FGFR2b stimulated with FGF7 (dark green) or FGF10 (burgundy). Data represent the mean ± SD of N = 3 compared with FGF10. P =< 0.05*, < 0.01**, < 0.001*** (Student's t-test).
E  Hierarchical clustering of the phosphorylated sites differentially quantified T47D stimulated with FGF7 (green) or FGF10 (burgundy). Specific clusters are highlighted with black lines. The intensity of phosphorylated sites is presented on the logarithmic scale with intensity below and above the mean colour-coded in blue and red, respectively.
F  Enriched terms in the selected clusters of E, cluster FGF10 (burgundy, top), cluster FGF7 (green, bottom).
G  Network of phosphorylated proteins belonging to "protein localization" in FGF10 clusters, based on STRING, visualized in Cytoscape, and colour-coded based on cell components. The squared shape represents phosphorylated proteins found in the database DISEASES. The recycling adaptors TTP and RCP are highlighted in grey.
H  Internalization (cytoplasm) and recycling (plasma membrane) of FGF10-stimulated HA-FGFR2b (green) transfected in BT549 stimulated for 0, 40 and 120 min. Cells were depleted or not of TTP or RCP by a pool of siRNAs. TRITC-Tf is a marker of recycling (red). Nuclei are stained in blue. *, cells with receptor recycled to the plasma membrane. Scale bar, 5 μm.
I  The presence (total), internalization (internalized) and recycling (cell surface) of transfected HA-FGFR2b in BT549 upon stimulation were quantified as in (Francavilla *et al*, 2016) and in the section "Quantification of the Recycling assay". Values represent the median ± SD of N = 3. Representative pictures upon FGF10 stimulation are shown in H.

separated the phosphoproteome to a greater extent than the duration of stimulation (Fig 1E and F, Appendix Fig S2H and I), confirming the uniqueness of ligand responses (Francavilla *et al*, 2013). Analysis of the FGF7- and FGF10-specific clusters showed enrichment of the GO term protein localization unique to FGF10 stimulation in both T47D and BT20 (Fig 1E and F, Appendix Fig S2H and I). In support of the idea that the FGF10 phosphoproteome regulates protein localization in breast cancer cells, we found that the FGF10 phosphorylated proteins were enriched in all cellular compartments including endosomes and that 63% of them were ascribed to human diseases, including breast cancer (Pletscher-Frankild *et al*, 2015; Fig 1G). Furthermore, the FGF10-regulated phosphoproteome in T47D and BT20 contained TTP and RCP (Fig 1G), known to regulate FGFR2b and EGFR recycling, respectively (Francavilla *et al*, 2013; Francavilla *et al*, 2016). Confocal microscopy confirmed that FGF10-induced FGFR2b recycling required TTP (Francavilla *et al*, 2013), but also RCP (Fig 1H and I, Appendix Fig S2J). Altogether, these data highlight a role for TTP and RCP in FGFR2b trafficking and early signalling specifically induced by the recycling ligand FGF10 in breast cancer cells.

Next, we studied how widely recycling affects signalling in breast cancer cells by assessing the contribution of three recycling FGFRs and of the recycling adaptors TTP and RCP to changes in the phosphoproteome, hereafter referred to as TPA2 and TPA3, respectively (Fig 1A, Datasets EV3–EV5). TPA2 compares the signalling downstream of recycling FGFRs in a defined genetic background. We overexpressed FGFR1c, 2b, or 4 in BT549 cells, which lack these FGFRs, and stimulated cells with Enkamin-E, FGF10, or FGF1 for 8 and 40 min., respectively (Fig 2A). It is known that each of these ligands induced recycling of the paired receptor (Haugsten *et al*, 2005; Francavilla *et al*, 2013; Francavilla *et al*, 2016) and we showed also ligand-dependent sustained signalling activation (Appendix Fig S3A and B). Hierarchical clustering of the 6402 phosphorylated sites from this high-quality dataset (Appendix Fig S3C–H) identified a cluster related to early signalling (8 min) and one associated with late endosomal signalling (40 min.) common to all the considered FGFR-ligand pairs

(Fig 2B). Refined analysis of this recycling receptor cluster revealed a signalling network of 866 proteins of which 38 were known trafficking proteins and 24, among which EGFR, TTP and RCP, were also identified by TPA1 (Figs 1 and 2C, and Appendix Figs S1–S3I). TPA3 analysed the FGF10-dependent phosphoproteome of T47D in the presence or absence of the recycling adaptors TTP or RCP (Fig 2D, Appendix Fig S3J). We identified 9569 phosphorylated sites and verified the high level of correlation between replicates of TPA3 (Appendix Fig Sl-O). Hierarchical clustering identified TTP- and RCP-specific clusters and a common recycling adaptor cluster (Fig 2E). The analysis of 113 proteins associated with the GO term establishment of localization in the latter cluster revealed 22 proteins already identified by TPA1, of which 6 were kinases, including EGFR (Figs 1 and 2F, and Appendix Figs S1, S2 and S3P). To identify key regulators of signalling downstream from FGFR recycling in breast cancer cells based on the multi-angle TPAs, we focused on phosphorylated proteins belonging to the GO term establishment of localization and prioritized the 22 phosphorylated proteins in common to the three TPAs (Fig 2G). This group of proteins included scaffolding proteins and the three protein kinases AKT, PAK1 and, strikingly, EGFR (Fig 2G). Assessment of EGFR-phosphorylated sites that were differentially regulated within each TPA highlighted that the phosphorylation of the non-catalytic threonine 693 (EGFR_T693) was uniquely associated with a recycling signature (40 min. upon stimulation) (Fig 2G). Therefore, the three quantitative TPAs developed to study FGFR recycling-dependent signalling integration in breast cancer cells (Fig 1A) unveiled a FGFR recycling-associated EGFR_T693 phosphorylation (Figs 1G, 2C, F, G and Datasets EV1–EV5). This finding suggests a hitherto unknown signalling interplay between FGFRs and EGFR in breast cancer cells. As EGFR_T693 phosphorylation is critical for EGF-induced EGFR internalization (Heisermann *et al*, 1990) and is induced by the EGFR recycling stimulus TGFα in a sustained manner (Francavilla *et al*, 2016), we hypothesized that EGFR_T693 phosphorylation may contribute towards FGFR outputs that depend on FGFR recycling.

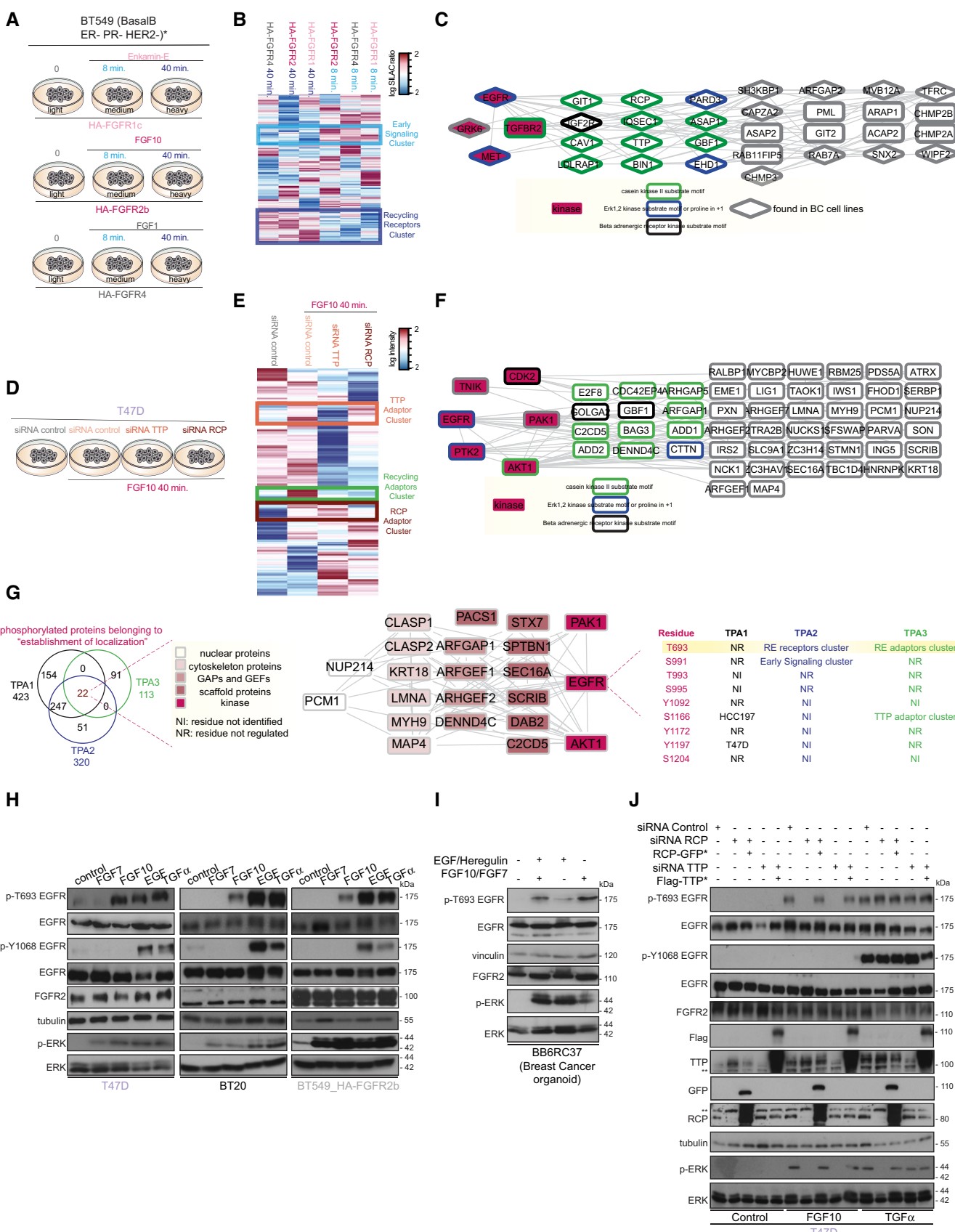

**Figure 2.**

**Figure 2. TPA2-3 unveil FGF10-dependent EGFR_T693 phosphorylation.**

A   Experimental design of TPA2.
B   Hierarchical clustering of the phosphorylated sites differentially quantified in BT549 transfected with HA-FGFR1c, 2b or 4 and stimulated for 8 or 40 min. with Enkamin-E, FGF10 or FGF1, respectively. Early signalling and recycling receptor clusters are highlighted in light and medium blue, respectively. The intensity of phosphorylated sites is presented on the logarithmic scale with intensity below and above the mean colour-coded in blue and red, respectively.
C   STRING-based and Cytoscape-visualized network of the phosphorylated proteins belonging to "endocytosis" (clusters medium blue in b). The diamond shape represents phosphorylated proteins found in TPA1. Kinases are highlighted in burgundy. The border is colour-coded based on the substrate motifs.
D   Experimental design of TPA3.
E   Hierarchical clustering of the phosphorylated sites differentially quantified in T47D stimulated with FGF10 and depleted or not of TTP or RCP. The clusters for TTP adaptors, RCP adaptors or recycling adaptors are highlighted in orange, brown and light green, respectively. The intensity of phosphorylated sites is presented on the logarithmic scale with intensity below and above the mean colour-coded in blue and red, respectively.
F   STRING-based and Cytoscape-visualized network of phosphorylated proteins belonging to the recycling adaptor cluster (light green in E) and found in TPA1. Kinases are highlighted in burgundy. The border is colour-coded based on the substrate motifs.
G   Venn diagram showing the phosphorylated proteins belonging to "establishment of localization" identified in TPA1-3 (left). STRING-based and Cytoscape-visualized network of the 22 proteins identified by the 3 TPAs (centre). Phosphorylated sites quantified on EGFR (right). T693 is highlighted in yellow.
H–J   Lysates from (H) T47D, BT20 and HA-FGFR2b-transfected BT549 stimulated or not with FGF7, FGF10, EGF and TGFα for 40 min; (I) breast cancer organoid cultured from the PDX tumour BB6RC37 and grown for the last 24 h as indicated; (J) control or 40 min. FGF10- or TGFα-stimulated T47D left untreated or depleted of TTP, followed or not by transfection with siRNA-resistant Flag-TTP (Flag-TTP*) or depleted of RCP followed or not by transfection with siRNA-resistant RCP-GFP (RCP-GFP*) were immunoblotted with the indicated antibodies. **, non-specific band (J).

## EGFR_T693 phosphorylation is FGFR2b ligand-, recycling- and activation-dependent

To verify the potential interplay between FGFR and EGFR, we first validated EGFR_T693 phosphorylation in T47D, BT20, HA-FGFR2b-BT549 and a breast cancer organoid grown from a TNBC patient-derived xenograft (PDX) tumour (Eyre *et al*, 2016). FGF10, but not FGF7, induced EGFR phosphorylation on T693, whilst leaving the catalytic residue tyrosine 1068 (Y1068) unaltered at 40-min. stimulation (Maennling *et al*, 2019). By contrast, the EGFR ligands EGF and TGFα induced the phosphorylation of both T693 and Y1068, as shown previously (Ceresa & Peterson, 2014) (Fig 2H and I). Therefore, sole phosphorylation of EGFR at T693 is FGF10/FGFR2b-specific.

To confirm that FGF10-induced FGFR2b recycling was involved in EGFR_T693 phosphorylation, we depleted TTP or RCP and stimulated T47D and BT20 with FGF10 for 40 min. This resulted in decreased EGFR_T693 phosphorylation and ERK activation upon depletion of either recycling adaptor (Appendix Fig S4A and B). Overexpressing siRNA-resistant TTP or RCP restored both EGFR_T693 phosphorylation and ERK phosphorylation following FGF10 stimulation (Fig 2J). Furthermore, we observed a peak in EGFR_T693 phosphorylation at 40 min. post-stimulation with FGF10 when FGFR2b was present in recycling endosomes (Fig 1B–I, Appendix Fig S4C). Finally, FGF10-mediated phosphorylation of EGFR_T693 decreased when FGFR2b trafficking was inhibited by dominant-negative Rab11 (preventing recycling) or dominant-negative dynamin (preventing internalization) with no discernible alterations in ERK activation (Appendix Fig S4D-G). As TGFα or EGF-induced EGFR_T693 phosphorylation occurs regardless of length of stimulation, or trafficking inhibition (Fig 2J, Appendix Fig S4), we concluded that EGFR_T693 phosphorylation is independent of EGFR recycling, but FGFR2b-induced phosphorylation of EGFR on T693 requires recycling upon FGF10 stimulation.

We next verified whether the kinase activity of FGFR was required for EGFR_T693 phosphorylation. Enkamin-E, FGF10 and FGF1 induced EGFR_T693 phosphorylation in cells expressing their cognate receptors, but this was suppressed by the FGFR inhibitor PD173074 (Pardo *et al*, 2009) (Fig 3A and B). Similarly,

FGF10-dependent EGFR_T693 phosphorylation decreased in cells transfected with the catalytically inactive HA-FGFR2b_Y656F/Y657F (Francavilla *et al*, 2013) (Fig 3C). Furthermore, FGF10, but not TGFα, required FGFR activation to induce EGFR_T693 phosphorylation and ERK activation in breast cancer cells expressing endogenous FGFR2b (Fig 3D). Conversely, TGFα-, but not FGF10-induced EGFR_T693 phosphorylation was blocked by the EGFR inhibitor AG1478 (Han *et al*, 1996) (Fig 3D). These data suggest that EGFR_T693 phosphorylation downstream from FGFR ligands depends on FGFR but not EGFR activation. As a conserved proline follows T693 on EGFR and TPA2-3 uncovered an ERK-kinase-substrate (or proline in +1) motif encompassing the phosphorylated EGFR (Figs 2 C–F and 3E), we treated T47D and BT20 with MEK inhibitors, U0126 and MEK162 (Cheng & Tian, 2017), or the p38 kinase inhibitor BMS582949 (Emami *et al*, 2015) (Fig 3A). Both MEK inhibitors blocked ERK phosphorylation in cells stimulated with FGF10 and TGFα, and EGFR_T693 phosphorylation was simultaneously decreased upon FGF10 treatment, and to lesser extent upon TGFα, whereas p38 inhibition had no effect (Fig 3F–G). Therefore, EGFR_T693 phosphorylation depended on MEK-ERK, but not p38, signalling upon FGF10 stimulation.

In conclusion, FGF ligands which induce the recycling of their paired FGFR receptor increase EGFR_T693 phosphorylation via FGFR and ERK signalling, independent of EGFR or p38 activity.

## FGF10 primes EGFR responses

To explore the consequences of the FGFR and EGFR interplay, we first tested the effect of FGF10 on EGFR functions. FGF10 stimulation did not alter the levels of EGFR, which decrease over time upon EGF—and to a lesser extent TGFα—stimulation (Francavilla *et al*, 2016) (Fig 4A). However, if T47D cells were pre-treated for 40 min with FGF10, followed by stimulation with EGF for different time points, we observed an increase in EGFR abundance and ERK activation relative to cells not pre-treated with FGF10 for 120 min (Fig 4 B). FGF10 pre-treatment did not have the same effect on EGFR stabilization from the recycling stimulus TGFα at 120 min and a less pronounced effect on ERK phosphorylation at any time point. These data suggest that FGF10 pre-treatment alters EGF signalling to increase stability of EGFR at 120 min resulting in higher levels of

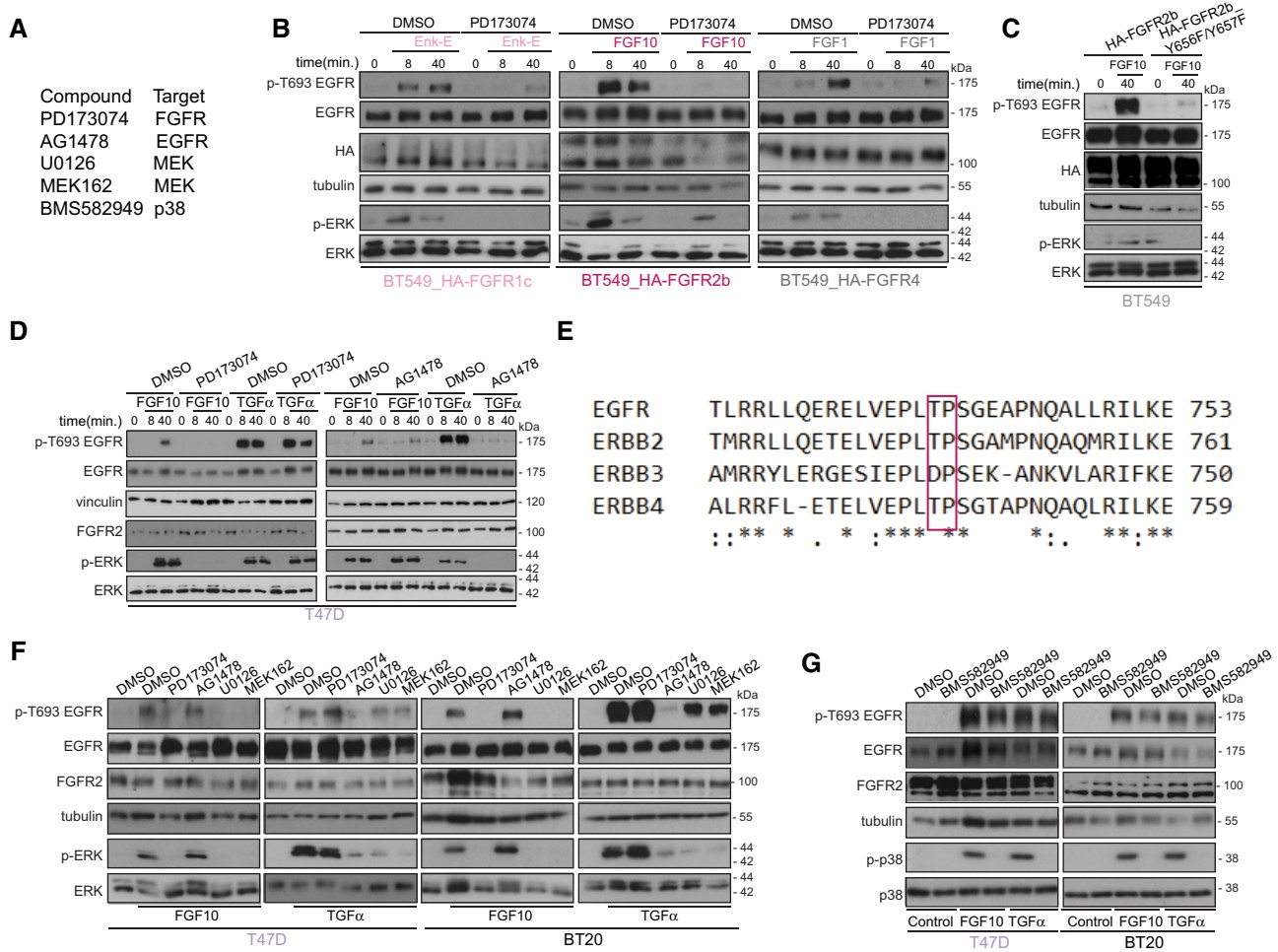

**Figure 3. FGF10-dependent EGFR_T693 phosphorylation requires FGFR and ERK.**

A   List of compounds and their targets.

B–D   Lysates from BT549 transfected with HA-FGFR1c, 2b, or 4 and stimulated for 8 or 40 min with Enkamin-E, FGF10 or FGF1, respectively, followed by treatment with either DMSO or the FGFR inhibitor PD173074 (B); BT549 transfected with HA-FGFR2b or HA-FGFR2b_Y656F/Y657F and stimulated with FGF10 for 40 min. (C); T47D treated with DMSO, PD173074 or the EGFR inhibitor AG1478 and stimulated or not with either FGF10 or TGFα for 0, 8 and 40 min. (D) were immunoblotted with the indicated antibodies.

E   Protein sequences surrounding T693 (based on UniProt P00533) of members of the ErbB family were aligned using CLUSTAL O (version 1.2.4). The red box indicates that the amino acid T is followed by a conserved proline. Asterisks below the sequences indicate identical amino acid residues; double dots indicate conserved amino acid residues; single dots indicate semi-conserved substitutions.

F, G   Lysates from T47D or BT20 treated with DMSO, PD173074, AG1478 or the MEK inhibitors U0126 and MEK162 and stimulated or not with either FGF10 or TGFα (F); T47D or BT20 treated with DMSO, or the p38 inhibitor BMS582949 and stimulated or not with either FGF10 or TGFα (G) were immunoblotted with the indicated antibodies.

ERK activation. FGFR2 levels remained high with all treatment conditions; however 40 min., FGF10 pre-treatment did alter the dynamics of FGFR2b downstream of both EGF and TGFα stimulation. The effect of FGF10 pre-treatment on EGFR dynamics in EGF-stimulated cells depended on both FGFR and ERK activities (Fig 4C and D). When EGFR was stabilized with AG1468 (Gan *et al*, 2007), the stability of the EGFR following pre-treatment was in line with FGF10 40-min stimulation, whereas ERK activation decreased (Fig 4 E). Therefore, FGF10 pre-treatment increases the total levels of EGFR after EGF stimulation, and this resulted in sustained ERK phosphorylation, an effect dependent on the activation of the FGFR-ERK signalling axis. Correspondingly, FGF10 pre-treatment followed

by EGF stimulation for 4h induced the highest expression of ERK late target genes (Uhlitz *et al*, 2017) (Fig 4F), which may suggest enhanced cell cycle progression (Sharrocks, 2006). This was confirmed by increased EdU incorporation in T47D and BT20 stimulated with EGF upon pre-treatment with FGF10 (Fig 4G, Appendix Fig S5A). The use of specific inhibitors indicated that FGF10 significantly increased EGF-dependent cell cycle progression in an FGFR-, ERK-, and EGFR-dependent manner (Fig 4G and H and Appendix Fig S5).

In conclusion, FGF10 pre-treatment stabilizes EGF-stimulated EGFR via FGFR-ERK signalling. Elevated ERK phosphorylation, increased expression of ERK late target genes, and enhanced cell

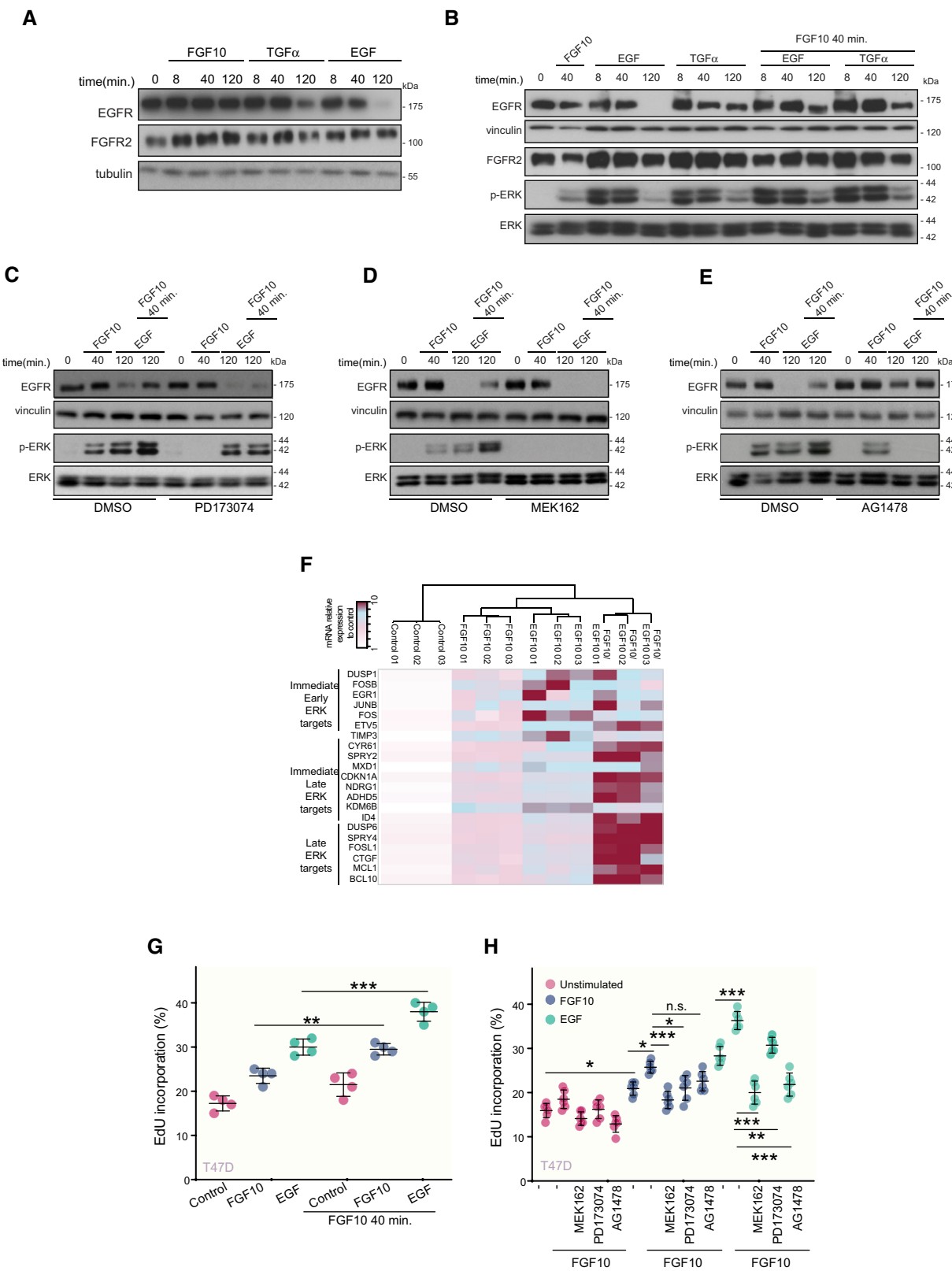

**Figure 4.**

**Figure 4. FGF10 primes EGFR responses.**

A–E   Lysates from T47D stimulated or not with FGF10, EGF or TGFα for different time periods (A); pre-treated or not with FGF10 for 40 min. and either stimulated or not with FGF10, EGF or TGFα for different time periods (B) or treated with PD173074 (C), MEK162 (D), AG1478 (E) before stimulation were immunoblotted with the indicated antibodies.

F   Heatmap of the mRNA relative expression of the indicated ERK targets compared with control and quantified by qPCR. $N = 3$. The minimum and maximum fold-induction is colour-coded in white and burgundy, respectively.

G, H   Percentage of EdU incorporation in T47D pre-treated or not with FGF10 for 40 min. and stimulated or not with FGF10 or EGF (G) or incubated with MEK162, PD173074 or AG1478 before pre-treatment (H). $N = 6$. $P =< 0.05^*, < 0.01^{**}, < 0.001^{***}$ (one-way ANOVA with Tukey test).

cycle progression suggest that FGF10 pre-treatment primes EGF responses and confirms a functional interplay between FGFR and EGFR signalling.

**FGF10-dependent EGFR_T693 phosphorylation in the recycling endosome reciprocate priming of FGFR2b outputs**

The FGFR/EGFR interplay was also verified by uncovering the co-localization of FGFR2b with EGFR in recycling endosomes upon 40-min. stimulation with FGF10, but not with TGFα (Fig 5A and B, Appendix Fig S6A and B). Intriguingly, EGFR was phosphorylated on T693 in recycling endosomes at 40, but not at 20 min, in response to FGF10 (Fig 5C and D, Appendix Fig S6C). We therefore hypothesized that the recycling endosomes may form the interface for FGFR and EGFR signal integration, perhaps involving physical interaction of the receptors, and that EGFR_T693 phosphorylation could affect FGFR2b trafficking. We assessed FGFR2b/EGFR co-localization and co-immunoprecipitation in T47D cells depleted of endogenous EGFR and transfected with either EGFR wild-type (wt) or the T693A mutant (T693A)—which cannot be phosphorylated at residue T693—upon a time-course stimulation with FGF10 (Fig 5, Appendix Fig S6, Appendix Fig S7A and B). Under non-stimulated conditions, FGFR2b co-localized at the plasma membrane and co-immunoprecipitated with both wt and T693A EGFR (Fig 5E–G, Appendix Fig S6D–F). At 20-min. stimulation with FGF10, FGFR2b was detected in the recycling endosomes in both wt- and T693A-expressing cells, but it failed to interact or localize with either EGFR as both were still located at the plasma membrane (Fig 5E–F, Figs EV1 and EV2, Appendix Fig S6G). Therefore, FGFR2b traffics to the recycling endosomes before interacting with EGFR. We also observed that FGFR2b localization to the recycling endosomes in FGF10 stimulated wt-expressing cells for 40 min. was lost in the presence of T693A. At this time point, FGFR2b co-immunoprecipitated with both wt and T693A (Fig 5G), but co-localized with wt in recycling endosomes and with T693A at the plasma membrane. These data imply that FGFR2b trafficking is altered in cells expressing T693A EGFR. Interestingly, RCP failed to interact with FGFR2b in T693A-expressing cells stimulated for 40 min. with FGF10 (Fig 5G, Appendix Fig S6F, suggesting that RCP and EGFR interact with FGFR2b in recycling endosomes in a T693 phosphorylation-dependent manner. Finally, at 60-min stimulation with FGF10, we detected FGFR2b and EGFR at the plasma membrane in both wt- and T693A-expressing cells (Fig 5E–G, Appendix Fig S6F–G). TGFα stimulation of T693A-expressing cells confirmed that T693 phosphorylation regulates EGFR internalization (Heisermann *et al*, 1990), as the T693A receptor was unable to traffic from the plasma membrane under any of the tested conditions (Fig 5E and F). Using GFP-Rab11-APEX2, which did not affect

FGFR2b trafficking (Appendix Fig S6H), we assessed at which time point the majority of EGFR phosphorylated on T693 was detected in proximity to the recycling endosomes following FGF10 stimulation. In agreement with the confocal imaging, we found that EGFR phosphorylation on T693 accumulated at the recycling endosomes between 20- and 40-min stimulation, when FGFR2b was detected in recycling endosomes together with RCP (Appendix Fig S6G–I, Fig 5). Altogether, these data suggest that EGFR_T693 phosphorylation dynamically regulates FGFR2b trafficking after the formation of an FGFR2b/EGFR/RCP complex in the recycling endosomes. Indeed, in T693A-expressing and in cells treated with FGFR and ERK inhibitors, but not EGFR or p38 inhibitors, there is less intracellular FGFR2b and increased receptor at the cell surface at 40-min. stimulation (Fig 5H, Appendix Fig S6J and K). Therefore, FGF10-dependent EGFR_T693 phosphorylation via FGFR and ERK activation (Fig 3) plays a role in the spatio-temporal regulation of FGFR2b trafficking.

To test the cellular impact of FGFR/EGFR signalling integration by quantifying whether EGFR_T693 phosphorylation affects FGF10 signalling downstream from FGFR2b, we compared the phosphoproteome of T47D expressing either wt or T693A EGFR upon stimulation with FGF10 for 40 or 60 min (Fig 6A, Dataset EV6). We confirmed the absence of T693 phosphorylation of T693A EGFR, whilst other EGFR residues and ERK were phosphorylated in both wt- and T693A-expressing cells (Appendix Fig S7A and B). The reproducibility of this dataset was consistent with the previous ones (Appendix Figs S1–S3, S7C–G, Datasets EV1–EV6). Hierarchical clustering of the 6485 identified phosphorylated sites revealed 4 clusters. Three clusters grouped sites whose phosphorylation increased in T693A- compared with wt-expressing cells. These clusters (plasma membrane response, acquired response, and late response) were enriched for general cellular processes (Fig 6B and C). The fourth cluster (T693 phosphorylation-dependent response) represented phosphorylated sites dependent on EGFR_T693 phosphorylation downstream from FGF10 signalling (Fig 6B and C). Of the 102 phosphorylated sites identified on the 53 kinases within all the four clusters, 10 were known regulatory sites (Fig 6D, Datasets EV6 and EV7). More specifically, FGF10-stimulated T693A-expressing cells showed decreased phosphorylation of proteins belonging to the GO term cell cycle, including the activating T161 site on the cell cycle regulator cyclin-dependent kinase 1 (CDK1) (Coulonval *et al*, 2011) (Fig 6D, Appendix Fig S7H, Dataset EV7). We confirmed that FGF10 induced CDK1_T161 phosphorylation in wt-, but not T693A-expressing T47D and BT20 cells (Fig 6E–F). Furthermore, FGF10-mediated cell cycle progression decreased in T693A-expressing cells, an effect due to impaired EGFR_T693 phosphorylation, as the total level of EGFR wt or T693A did not alter over time (Fig 7A and B, Appendix Fig S7I). Therefore, FGF10-

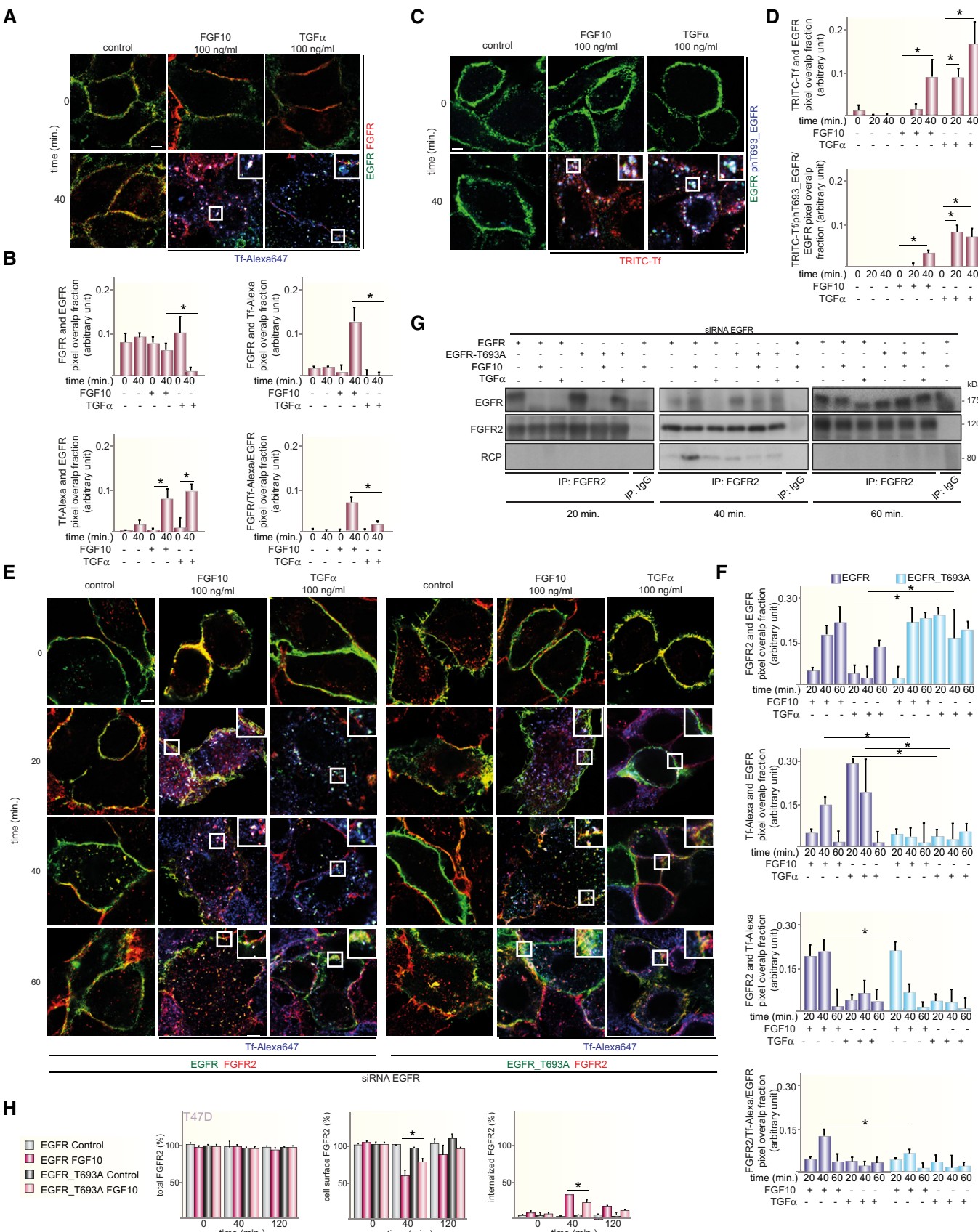

**Figure 5.**

**Figure 5.  EGFR_T693 phosphorylation regulates FGFR2b recycling.**

A   Co-localization of FGFR2 (red), EGFR (green) and the recycling marker Tf (blue) in T47D stimulated or not with FGF10 or TGFα for 40 min. Scale bar, 5 μm.

B   Red and green pixels overlap fraction (above, left) representing the co-localization of FGFR2 with EGFR; red and far-red pixels overlap fraction (above, right) representing the co-localization of FGFR2 with the recycling marker Tf; far-red and green pixels overlap fraction (below, left) representing the co-localization of EGFR with the recycling marker Tf; green, red and far-red pixels overlap fraction (below, right) representing the co-localization of FGFR2, EGFR, and the recycling marker Tf in T47D stimulated for 40 min. Values represent the median ± SD of at least 3 independent experiments. Representative pictures are shown in A and Appendix Fig S6A. *, *P*-value<0.005 (Student's *t*-test).

C   Co-localization of EGFR (green), T693 phosphorylated EGFR (blue) and the recycling marker Tf (red) in T47D stimulated or not with FGF10 or TGFα for 40 min. Scale bar, 5 μm.

D   Red and green pixels overlap fraction (above) representing the co-localization of EGFR with the recycling marker Tf; green, red and far-red pixels overlap fraction (below) representing the co-localization of EGFR, phosphorylated EGFR and the recycling marker Tf in T47D stimulated for 40 min. Values represent the median ± SD of at least 3 independent experiments. Representative pictures are shown in C and Appendix Fig S6C. *P* < 0.005 (Student's *t*-test).

E   Co-localization of FGFR2 (red), EGFR (green) and the recycling marker Tf (blue) in T47D depleted of EGFR by siRNA followed by transfection with wt or T693A and stimulated or not with either FGF10 or TGFα for the indicated time periods. Scale bar, 5 μm.

F   Red and green pixels overlap fraction representing the co-localization of FGFR2 with EGFR; far-red and green pixels overlap fraction representing the co-localization of EGFR with the recycling marker Tf; red and far-red pixels overlap fraction representing the co-localization of FGFR2 with the recycling marker Tf; green, red and far-red pixels overlap fraction representing the co-localization of FGFR2, EGFR and the recycling marker Tf in cells stimulated for 20, 40 or 60 min. with FGF10 or TGFα. Values represent the median ± SD of at least 3 independent experiments. Representative pictures are shown in E. *P* < 0.005 (Student's *t*-test).

G   Lysates from T47D depleted of EGFR by siRNA followed by transfection with siRNA-resistant wt or T693A and stimulated or not with either FGF10 or TGFα for the indicated time periods were used for immunoprecipitation of FGFR2 and then immunoblotted with the indicated antibodies. The inputs are shown in Appendix Fig S6F.

H   The presence (total), internalization (internalized) and recycling (cell surface) of FGFR2 in T47D depleted of EGFR by siRNA followed by transfection with wt or T693A and stimulated with FGF10 for different time periods were quantified as described (Francavilla *et al*, 2016) and in the section 'Quantification of the Recycling Assay'. Briefly, we assessed approximately 100 cells per condition and expressed the results as the percentage of receptor-positive cells over total cells (corresponding to DAPI-stained nuclei) and referred to the values obtained at time zero. Values represent the median ± SD of *N* = 3. *P* < 0.005 (Student's *t*-test).

mediated FGFR2b recycling, CDK1 phosphorylation, and the level of cell cycle progression all depend on EGFR_T693 phosphorylation. Finally, FGF10-dependent EGFR_T693 phosphorylation plays a crucial role in FGF10 priming of EGFR responses, as shown by decreased EdU incorporation in T693A-expressing T47D and BT20 cells stimulated with FGF10 for 40 min prior to TGFα treatment (Fig 7C and D). We have previously shown that FGF10-mediated FGFR2b recycling regulates cell migration (Francavilla *et al*, 2013) and RTK recycling is known to increase cell motility (Crupi *et al*, 2020). As FGFR2b recycling was impaired in T693A-expressing cells (Fig 5), we then investigated whether EGFR_T693 phosphorylation was required for cell invasion. Surprisingly, FGF10 stimulation increased cell invasion in both wt- and T693A-expressing cells (Fig 7E and F), implying that EGFR_T693 phosphorylation is not important for FGF10-induced cell invasion. In conclusion, our data showed that recycling endosomes facilitate EGFR_T693 phosphorylation induced by FGF10 and that T693 phosphorylation is required for the full response to FGF10 and for FGF10 to prime EGF responses (Fig 7G, Fig EV3).

Overall, the "reciprocal priming" between FGFR and EGFR is a novel mechanism to coordinate the trafficking and the signalling outputs of these two RTKs in breast cancer cells.

# Discussion

This study shows that FGFR activation primes EGF-mediated responses in breast cancer cells whilst inducing EGFR_T693 phosphorylation from the recycling endosomes. Such phosphorylation events in turn modify the FGFR responses. This reciprocal priming between FGFR and EGFR from the recycling endosomes alters the dynamics of recycling and enhances cell cycle progression, but not cell invasion, downstream from both RTKs. It represents the first early, selective and multi-functional mechanism of RTK signalling integration which drives long-term outputs. In contrast to the known RTK cross-talk, where the inhibition of a dominant RTK may result in the compensatory recruitment of signalling molecules to a second RTK (Cao, 2016), reciprocal priming does not occur sequentially, rather simultaneously during the entry route of each RTK into the cytoplasm. This is an efficient way to rapidly change cell behaviour in response to the presence of a combination of ligands in the cell environment. Based on our comprehensive resource which integrates four quantitative phosphoproteomics datasets, we suggest that several recycling and signalling factors identified in this study may play a role in ensuring the co-localization of RTKs in the same cellular compartment. TPAs can be also explored by the community to pinpoint the molecular determinants of RTK recycling-dependent signalling in breast cancer cells. We therefore envision that reciprocal priming discovered in this study is unlikely to be restricted to FGFR and EGFR, opening up an exciting and novel avenue of RTK biology to be investigated. For instance, it remains to be determined whether FGFs induce the phosphorylation of threonine residues via ERK activity also on other RTKs. The discovery of a similarly phosphorylated peptide on c-Met depending on FGFRs recycling (Fig 2 C, Dataset EV4) and on other members of the EGFR family (Fig 3 E) suggests the presence of a network among RTKs in the recycling endosomes.

Besides regulating EGFR trafficking (Heisermann *et al*, 1990), T693 phosphorylation is a highly conserved residue (see response to referees in Review Process File available online) and involved in the response to stress or to the anti-tumour agent cisplatin via p38 activation (Winograd-Katz & Levitzki, 2006; Zwang & Yarden, 2006). Here, we show that FGFR ligands can induce EGFR_T693 phosphorylation and that T693 phosphorylation is required for the full activation of FGF10 responses and for FGF10 priming of EGFR outputs. This increases the repertoire of stimuli, including tumour necrosis factor-α (TNF-α) (Singhirunnusorn *et al*, 2007) or the Eph family

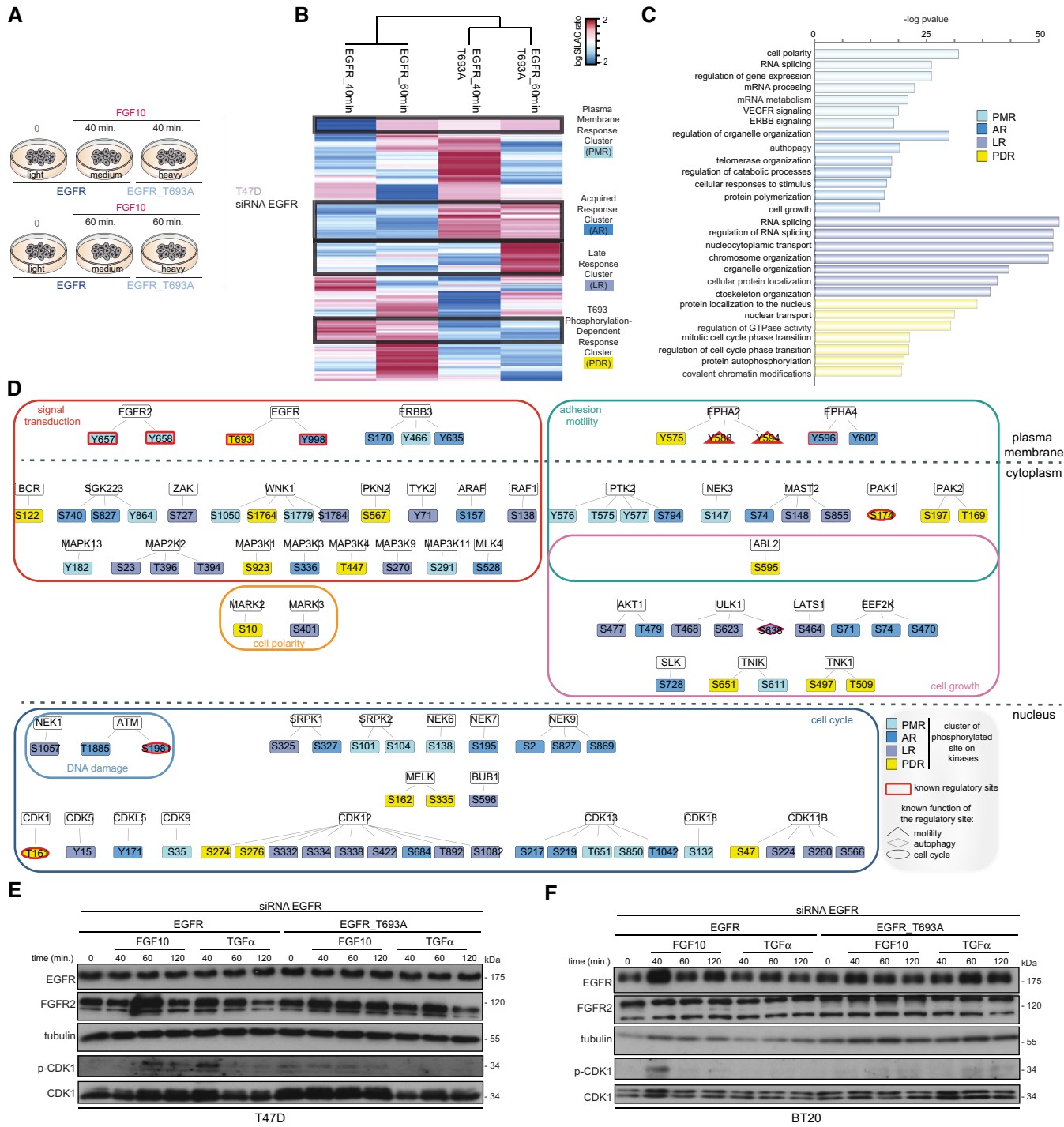

**Figure 6. EGFR_T693 phosphorylation controls CDK1 phosphorylation.**

A Experimental design of MS-based quantitative phosphoproteomics analysis of wt- and T693A-expressing T47D cells stimulated with FGF10 for 40 or 60 min.

B Hierarchical clustering of the phosphorylated sites differentially quantified in wt- and T693A-expressing T47D stimulated or not with FGF10 for 40 or 60 min. Four clusters for plasma membrane response (PMR), acquired response (AR), late response (LR), and T693 phosphorylation-dependent response (PDR) are highlighted with black lines and colour-coded on the right with blue, medium or dark blue and yellow, respectively. The intensity of phosphorylated sites is presented on the logarithmic scale with intensity below and above the mean colour-coded in blue and red, respectively.

C KEGG pathways enriched in each cluster.

D STRING-based and Cytoscape-visualized network of phosphorylated kinases colour-coded based on clusters shown in Fig 6C. The border of known regulatory sites is coloured in burgundy. The shape depends on the known function of the regulatory site.

E, F Lysates from T47D (E) and BT20 (F) depleted of EGFR, transfected with wt or T693A, and stimulated or not with either FGF10 or TGFα for the indicated time intervals were immunoblotted with the indicated antibodies.

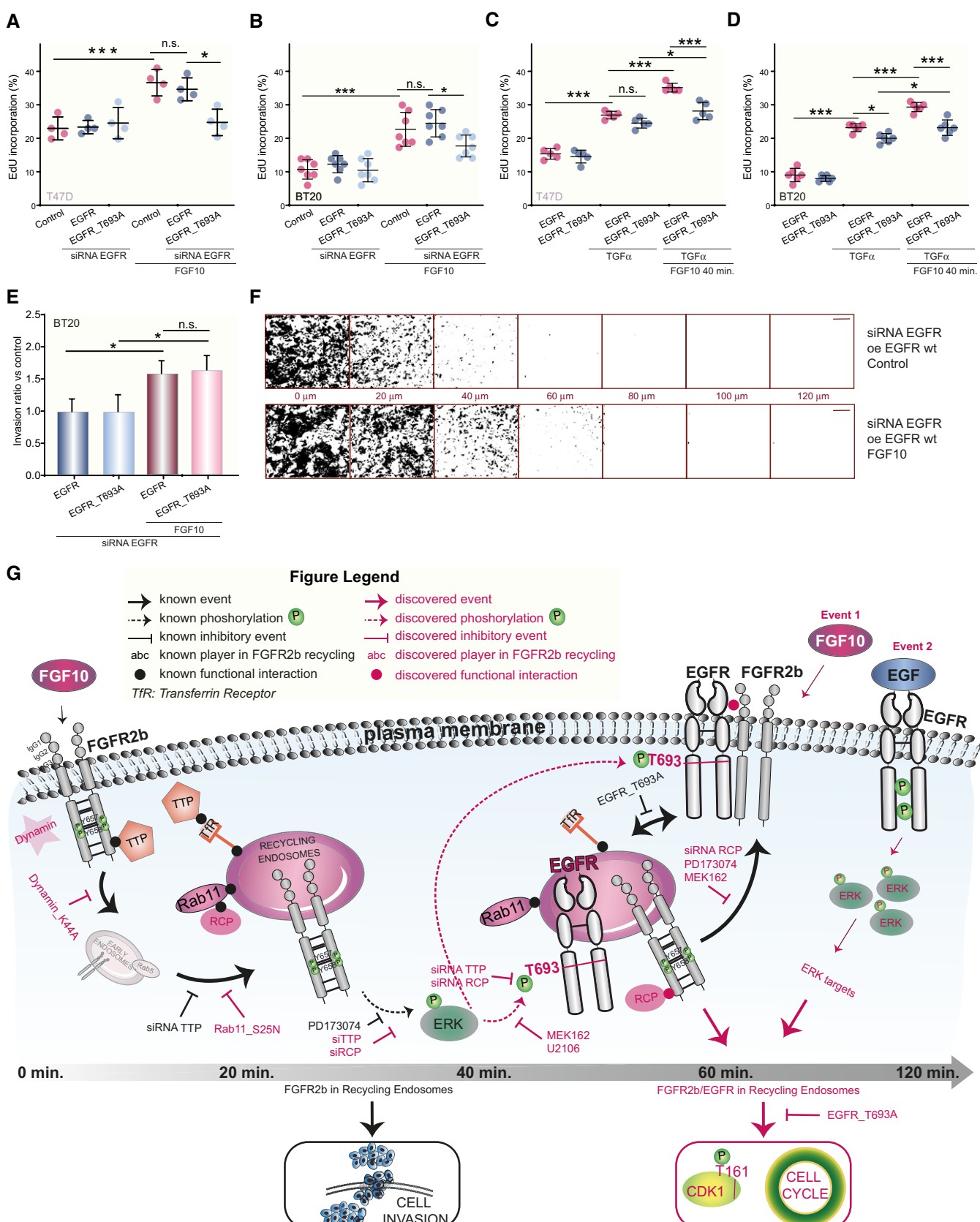

**Figure 7.**

**Figure 7. FGF10-induced cell cycle progression depends on EGFR_T693 phosphorylation.**

A–D  Percentage of EdU incorporation in T47D (A, C) or BT20 (B, D) depleted of EGFR or not, transfected with wt or T693A, and stimulated or not with FGF10 (A, B) or with TGFα (C, D) and pre-treated (C, D) or not (A, B) with FGF10 for 40 min. Values represent the median ± SD of N = 4. P =< 0.05*, < 0.01**, < 0.001*** (one-way ANOVA with Tukey test).
E  Relative invasion of wt- or T693A-transfected BT20 cells into fibronectin-supplemented collagen I was quantified as described in Material and Methods. Graph depicts mean ± SEM of N = 8. *P =< 0.05 (Student's t-test).
F  Representative images of E. Black indicates cells. Confocal depth is indicated between panels. Scale bar, 250 μm. oe, overexpression.
G  Model of reciprocal priming between FGFR and EGFR, based on this study.

(Stallaert *et al*, 2018), and the mechanisms which modulate EGFR activity with strong implications for RTK signalling integration. This concept is also supported by data showing that EGF-mediated phosphorylation of FGFR1 can be a route of RTK cooperation (Zakrzewska *et al*, 2013). Hence, further comprehensive studies are needed to fully understand how RTKs act in concert *in vivo* in response to the simultaneous presence of multiple ligands and how they ultimately regulate cell fate.

The dynamics of RTK trafficking is affected by several factors and in turns affects downstream signalling. For instance, EGFR recycles through recycling endosomes even in the absence of stimuli (Baumdick *et al*, 2015) and can be found in a subset of perinuclear compartments (Tomas *et al*, 2015). This potentially explains why the majority of EGFR_T693 phosphorylation is found at recycling endosomes upon FGF10 stimulation between 20 and 40 min, a time point when FGFR2b itself starts accumulating in the recycling endosomes. In turn, FGFR2b trafficking is deregulated when EGFR remains at the plasma membrane, e.g. in the absence of T693 phosphorylation, suggesting that the transient formation of the EGFR/FGFR2b complex on recycling endosomes is the key regulatory event for the correct *timing* of FGFR2b trafficking. One alternative model is that FGFR2b is internalized, traffics back to the plasma membrane where it retrieves EGFR, then reinternalizes (potentially recycling together EGFR), but the reinternalization step stalls without phosphorylation at T693. Another possibility is that FGFR2b phosphorylates the EGFR to prevent its recycling to the plasma membrane, thus explaining the change in EGFR distribution. This process should be replicated in FGF7-stimulated cells and would require further validation by high-resolution imaging of pools of FGFR2b and EGFR from the plasma membrane to recycling endosomes and back. The phosphorylation of EGFR might delay FGFR recycling back to the plasma membrane, thus allowing the formation of specific signalling complexes at the recycling endosomes. The kinases AKT and PAK1 identified by the three TPAs experiments would be interesting candidates to focus on to test this hypothesis. The alteration of the kinase landscape shown in T693A-expressing cells (Fig 6D) also confirms the idea that the EGFR phosphorylation may work as a scaffold for the recruitment of recycling machinery (e.g. RCP) and signalling partners to FGFR2b. The hypothesis of a multi-step regulation of FGFR2b—and possibly other RTKs—recycling is supported by our data on TTP and RCP. FGFR2b was not detected in recycling endosomes in TTP-depleted cells (Fig 1H); therefore, we propose that TTP is required for FGFR2b *entry* into recycling endosomes in epithelial cells (Fig 1I) (Francavilla *et al*, 2013). This would explain the lack of T693 phosphorylation and FGFR2b degradation in the absence of TTP. The third player in the complex regulation of FGFR2b recycling is RCP, which is bound to FGFR2b and EGFR in the recycling endosomes. As FGFR2b localized to recycling endosomes in RCP-depleted cells (Fig 1H), but it is partially degraded in this condition (Fig 1I), RCP may play a role in

FGFR2b *exit* from recycling endosomes. Therefore, RCP plays a temporally unique role in FGFR2b trafficking besides being a regulator of EGFR and integrin recycling (Caswell & Norman, 2008; Francavilla *et al*, 2016). We speculate on the presence of different populations of endosomes, either leading to recycling to plasma membrane (RCP-positive) or to receptor degradation (RCP-negative), implying that different families of RTKs regulate each other trafficking and signalling through a pool of adaptors on recycling endosomes. Future studies will reveal these adaptors upon different perturbations to build a more comprehensive regulatory network of RTK trafficking.

RTK recycling is known to control cellular responses, including proliferation, migration and invasion (Caswell & Norman, 2008; Francavilla *et al*, 2013; Francavilla *et al*, 2016). Here, we show that the multi-layered regulation of FGFR2b by ligand nature, trafficking route and EGFR priming fine-tunes downstream responses. Whereas cell migration and invasion require a signal from FGFR2b in recycling endosomes (Francavilla *et al*, 2013), the signal to fully promote the cell cycle occurs in a precise time window, when FGF10 primes FGFR2b and EGFR to recycling endosomes. As in the absence of the FGFR/EGFR reciprocal priming growth factors are less mitogenic but remain pro-migratory, we propose that recycling endosomes are dynamic signalling hubs that enable cells to coordinate cell cycle progression and cell invasion in response to multiple growth factors. Therefore, modulating RTK communication at early time windows might be an efficient way to re-direct cellular responses *in vivo*.

The simultaneous presence of RTKs has been described in breast cancer (Butti *et al*, 2018), where it might account for response to combined therapies (Issa *et al*, 2013), acquired resistance (Hanker *et al*, 2017) and epithelial cell-stroma communication (Wu *et al*, 2018). However, the concept of reciprocal priming has not been explored yet. It might have implications in TNBC, where both FGFRs and EGFR are highly expressed (Butti *et al*, 2018). Although we have not tested the effect of T693 phosphorylation in normal breast cells or in the stroma (Weber *et al*, 2005), this phosphorylated site may become a prognostic or predictive marker, if a correlation between T693 phosphorylation, clinical parameters and the response to combined EGFR/FGFR therapies is determined. This idea is supported by the detection of T693 phosphorylation in 50% of the TNBC patient-derived samples analysed in two independent phosphoproteomic datasets (Mertins *et al*, 2016; Huang *et al*, 2017). Importantly, reciprocal priming of RTKs may have implications when only one receptor is targeted as part of personalized therapies, where knock on effects to other pathways are not explored until resistance mechanisms have arisen (Tan *et al*, 2017). Targeting pan RTK trafficking (Porebska *et al*, 2018) and trafficking players, like TTP, RCP, and those identified by TPAs might open up novel therapeutic scenarios for treatment. Indeed, most of the recycling players identified here are mutated in breast cancer according to the

COSMIC database (Tate *et al,* 2019) and the importance of trafficking proteins in breast cancer is emerging (Wittkowski *et al,* 2018).

Understanding the extent of RTK regulation "in the right place at the right time" (Barrow-McGee & Kermorgant, 2014) is key for the integration of early signalling and long-term responses in cancer cells. Here, we unveiled a new form of RTK communication, a reciprocal priming coordinated from the recycling endosomes. Thus, the integration of TPAs offers a wealth of new candidates to investigate the functional consequences of trafficking-mediated signalling and has the potential to guide individualized treatment in cancer and other disease (Butti *et al,* 2018; Kobayashi *et al,* 2020).

# Material and Methods

## Materials availability

Further information and requests for resources and reagents should be directed to and will be fulfilled by Chiara Francavilla by email at chiara.francavilla@manchester.ac.uk.

## Reagents and Tools table

This information is provided in a separate Reagents and Tools Table.

| Reagent or Resource | Source | Identifier |
| --- | --- | --- |
| Antibodies | | |
| Rabbit anti Phospho-EGF Receptor (Tyr1068) Antibody | Cell Signaling Technology | 2234S |
| Mouse monoclonal Phospho-p38 MAPK (Thr180/Tyr182) (28B10) | Cell Signaling Technology | 9216S |
| Rabbit polyclonal CDK1 (phospho T161) | Abcam | ab47329-100ug |
| Rabbit polyclonal CDK1 | Abcam | ab131450-100ug |
| Mouse monoclonal FIP1/RCP antibody | Bio Techne | NBP2-20033 |
| Mouse monoclonal ERK 1/2 | Santa Cruz Biotechnology | sc-135900 |
| Mouse monoclonal γ-Tubulin | Sigma-Aldrich | T5326 |
| Mouse monoclonal Vinculin | Sigma Aldrich | V9264-200UL |
| Rabbit polyclonal pEGFR Thr669 | Cell Signaling Technology | 3056s |
| Rabbit monoclonal pEGFR Thr669 | Cell Signaling Technology | 8808s |
| Rabbit monoclonal p44/42 MAPK (Erk1/2) (137F5) | Cell Signaling Technology | 4695S |
| Mouse monoclonal GAPDH | Abcam | ab8245-100ug |
| Mouse monoclonal EGFR (Ab-1) | Merck | GR01L-100UG |
| Rabbit polyclonal EGFR | millipore | 06-847 |
| Rabbit monoclonal FGFR1 antibody D8E4 | Cell Signaling Technology | 9740 |
| Rabbit polyclonal SH3BP4 | Abcam PLC | ab106609-100ug |
| Rabbit monoclonal FGF Receptor 2 (D4L2V) | Cell Signaling Technology | 23328S |
| Rabbit monoclonal P38 | Cell Signaling Technology | 9212 |
| Rabbit monoclonal GFP | Cell Signaling Technology | 2956 |
| Peroxidase-AffiniPure F(ab')2 Fragment Goat Anti-Mouse IgG (H + L) (min X Hu, Bov, Hrs Sr Prot) | Stratech | 115-036-062-JIR-0.5ml |
| Peroxidase-AffiniPure F(ab')2 Fragment Goat Anti-Rabbit IgG (H + L) (min X Hu Sr Prot) | Stratech | 111-036-045-JIR-0.5m |
| Mouse monoclonal to EEA1 | BD Bioscience | 610457 |
| Goat anti-Rabbit IgG (H + L) Secondary Antibody, Alexa Fluor® 488 conjugate | Invitrogen | A11034 |
| Goat anti-Mouse IgG (H + L) Secondary Antibody, Alexa Fluor® 488 conjugate | Invitrogen | A11001 |
| Goat anti-Rabbit IgG (H + L) Secondary Antibody, Alexa Fluor® 568 conjugate | Invitrogen | A11011 |
| Donkey Anti-Mouse IgG (H + L) Secondary Antibody, Alexa Fluor® 647 conjugate | Invitrogen | A31571 |
| Donkey Anti-Rabbit IgG (H + L) Secondary Antibody, Alexa Fluor® 647 conjugate | Invitrogen | A31573 |

**Reagents and Tools table** (continued)

| Reagent or Resource | Source | Identifier |
| --- | --- | --- |
| Bacterial and Virus Strains | | |
| NEB® 10-beta Competent E. coli (High Efficiency) | New England Biolabs | Cat. No: C3019H |
| Biological Samples | | |
| Chemicals, Peptides, and Recombinant Proteins | | |
| Trypsin porcine pancreas (proteomics grade) | Sigma-Aldrich | T6567 |
| Lysyl Endopeptidase | FUJIFILM Wako Chemicals | 2541 |
| TiO beads "Titanspheres" | GL Sciences | 5020-75000 |
| Pre-cast gradient gel: Nu-PAGE 4-12% Bis-Tris Gel 1.0mm 10 well | Invitrogen | NP0321BOX |
| Sep-Pak Classic C18 cartridges | Waters | WAT051910 |
| Solid Phase Extraction Disk "Empore" C18 (Octadecyl) 3 M | Agilent Technologies | 2215 |
| Solid Phase Extraction Disk "Empore" C8 (Octyl) 3 M | Agilent Technologies | 2214 |
| L-ARGININE:HCL | Cambridge Isotope Laboratories | CLM-2265-H-0.25 |
| L-ARGININE:HCL | Cambridge Isotope Laboratories | CNLM-539-H-0.5 |
| L-ARGININE:HCL | Sigma-Aldrich | A6969 |
| L-LYSINE:2HCL | Cambridge Isotope Laboratories | DLM-2640-0.5 |
| L-LYSINE:2HCL | Cambridge Isotope Laboratories | CNLM-291-H-0.5 |
| L-LYSINE:2HCL | Sigma-Aldrich | L8662 |
| 2,5-Dihydroxybenzoic acid | Sigma-Aldrich | 85707 |
| RPMI 1640 Medium for SILAC | ThermoFisher Scientific | 88365 |
| TRIzol™ Reagent | ThermoFisher Scientific | Cat. No. 15596026 |
| DIHYDROETHIDIUM | Cambridge Bioscience | 12013-5mg-CAY |
| Hoechst 33342 | New England Biolabs | 4082S |
| Lipofectamine RNAiMAX Transfection Reagent | ThermoFisher Scientific | 10601435 |
| Lipofectamine Transfection Reagent | Life Technologies | 18324020 |
| FuGENE HD Transfection Reagent | Promega UK | E2311 |
| Sodium Pyruvate solution 100mM (100ml) | Sigma-Aldrich | S8636-100ML |
| Crystal violet solution | Sigma-Aldrich | V5265-250ML |
| Carestream Kodak BioMax MR Film | Kodak | Z350370-50EA |
| Xtra-Clear Flat 8-Strip Caps | Star labs | I1400-0900-C |
| 96-Well PCR Plate Non-Skirted Low Profile Natural | Star labs | E1403-0200-C |
| RPMI 1640 Medium Glutamax Supplement (500ml) | Gibco | 61870010 |
| ReliaPrep RNA Cell Miniprep System | NEB | Z6011 |
| Color Prestained Protein Standard Broad Range | NEB | P7712S |
| Prestained Protein Standard Broad Range | Sigma-Aldrich | SDS7B2 |
| PURELINK QUICK MINI | NEB? | K210010 |
| T4 DNA Ligase 20,000 u | NEB? | M0202S |
| DMEM High glucose HEPES w/o Glutamine and Sodium pyruvate | Sigma-Aldrich | D6171-6X500ML |
| DMEM AQ medium | Sigma-Aldrich | D0819-500ml |
| RPMI 1640 w/L-Glutamine-Bicarbonate | Sigma-Aldrich | R8758-6X500ML |
| Q5 High Fidelity 2x mastermix | NEB | M0492S |
| Nutrient Mix F12 HAM | Sigma-Aldrich | N6658-500ML |
| Human EGF (Animal Free) | PeproTech | AF-100-15-1000 |
| PRESTAINED MOLECULAR WEIGHT MARKER, MW 2 | Sigma-Aldrich | SDS7B2-1VL |
| MG132 | Fisher Sientific | 15465519 |
| HYPERFILM ECL 18X24CM | VWR International Ltd | 28-9068-37 |

**Reagents and Tools table**   (continued)

| Reagent or Resource | Source | Identifier |
| --- | --- | --- |
| Albumin, Bovine Fraction V (BSA), 100 Grams Cat No: A30075-100.0 | Melford Biolaboratories Ltd | A30075-100.0 |
| Bradford Reagent | Bio-Rad | 5000205 |
| Clarity ECL | Bio-Rad | 1705061 |
| GoScript Reverse Transcription Mix, Random Primers | Promega | A2801 |
| Pierce Protease Inhibitor Tablets-20 tablets | Life Tehnologies | A32963 |
| MEMBRANE PROTRAN 0,45uM NC 300MMX4 M | VWR | 10600002 |
| qPCRBIO SyGreen Mix Separate-ROX | pcr biosystems | PB20.14-50 |
| ProLong Diamond Antifade Mountant-2 mL | Life Technologies | P36965 |
| ExoSAP-IT | Life Technologies | 78250.40.ul |
| DMSO | Sigma-Aldrich | 276855-250ml |
| ACETONITRILE | VWR International Ltd | 1.00030.2500 |
| HEPARIN SODIUM CELL CULTURE TESTED | Sigma-Aldrich | H3149-100KU |
| Escort IV | SLS | L3287-1ML |
| Penicillin-Streptomycin (10,000 U/mL) | Life Technologies Ltd | 15140122 |
| Human EGF | Sigma-Aldrich | E9644-.2MG |
| Human TGFα | Pepro Tech Limited | 100-16A |
| Human FGF1 | Pepro Tech Limited | 100-17A-50 |
| Human FGF7 | Francavilla *et al* 2013 | PI: Prof Olsen |
| Human FGF3 | Bio Techne | 1206-F3-025 |
| Human FGF10 | Francavilla *et al* 2013 | PI: Prof Olsen |
| Enkamin-E | Pepro Tech Limited | A14-529EP |
| PD173074 | Selleckchem | S1264 |
| AG1478 | Cell Signalling Technologies | 9842 |
| U2106 | Cell Signalling Technologies | 9903 |
| MEK162 | APEXBIO | A1947 |
| BMS582949 | Selleck Chem | S8124 |
| Collagen I, HC, Rat Tail, 100 mg | Corning | 354249 |
| FIBRONECTIN FROM BOVINE PLASMA | Sigma | F1141-1MG |
| DMEM powder, high glucose | Thermo Fisher | 52100021 |
| Fetal Bovine Serum, South American origin | Life Technologies | 10270106 |
| TW PC MEMBRANE,6.5MM,8.0UM Transwell Inserts | Sigma Aldrich | CLS3422-48EA |
| Calcein AM cell permanent Dye | Fisher Scientific | C1430 |
| Glacial Acetic Acid (HPLC Grade) | Fisher Scientific UK | 10060000 |
| Formic Acid (HPLC Grade) | Sigma-Aldrich | 5438040250 |
| Trifluoracetic Acid (Spectroscopy Grade) | Sigma-Aldrich | 1082621000 |
| Dispase | Stem Cell Technologies | 7913 |
| Matrigel | Corning | 354230 |
| DAPI (4',6-Diamidino-2-Phenylindole, Dihydrochloride) | Life Technology | D1306 |
| Transferrin From Human Serum, Alexa Fluor™ 647 Conjugate | Invitrogen | T23366 |
| Transferrin From Human Serum, Tetramethylrhodamine Conjugate | Invitrogen | T2872 |
| Critical Commercial Assays | | |
| Click-iT EdU Alexa Fluor 488 Imaging Kit-1 kit | Life Technologies | C10337 |
| Venor®GeM Classic Mycoplasma PCR Detection Kit(100 tests) | Cambridge Bioscience | 11-1100 |
| ProtoScript; II First Strand cDNA Synthesis Kit | New England Biolabs | E6560L |
| ReliaPrep RNA Cell Miniprep System | Promega | Z6011 |

**Reagents and Tools table**  (continued)

| Reagent or Resource | Source | Identifier |
|---|---|---|
| Tumor dissociation kit | Miltenyi Biotec | 130-095-929 |
| Isolate II PCR and Gel kit | Bioline | BIO-52059 |
| Isolate II plasmid mini kit | Bioline | BIO-52056 |
| Deposited Data | | |
| Raw data (MS) | This paper | http://proteomecentral.proteomexchange.org/cgi/GetDataset (dataset identifier PXD018184) |
| Experimental Models: Cell Lines | | |
| MCF-7 | LGC ATCC® | HTB-22 |
| MDA-MB-415 | LGC ATCC® | HTB-24 |
| BT20 | LGC ATCC® | HTB-19 |
| HCC1937 | LGC ATCC® | CRL-2336 |
| T47D | LGC ATCC® | HTB-133 |
| BT549 | LGC ATCC® | HTB-122 |
| Experimental Models: Organisms/Strains | | |
| BB6RC37 | Eyre *et al* (2016) | PI: R. Clarke |
| Oligonucleotides | | |
| SIRNA UNIV NEGATIVE CONTROL #2 | Sigma-Aldrich | SIC002 |
| GGAGAUGAAAGUGUCAGCCGAGAUA | Invitrogen | SH3BP4HSS119149 |
| CCCAGGAUCUCAAGGUCUGUAUGUU | Invitrogen | SH3BP4HSS119150 |
| CCUGAUUGACCUGAGCGAAGGGUUU | Invitrogen | SH3BP4HSS119151 |
| GGUCCUCAAACAGAAGGAAACGAUA | Invitrogen | RAB11FIP1HSS149439 |
| GAAGACUACAUUGACAACCUGCUUG | Invitrogen | RAB11FIP1HSS149440 |
| UCCGCAUCCCGACUCAGGUUGGCAA | Invitrogen | RAB11FIP1HSS149441 |
| CGGAAUAGGUAUUGGUGAAUUUAAA | Invitrogen | EGFRHSS176346 (G01) |
| CCUAUGCCUUAGCAGUCUUAUCUAA | Invitrogen | EGFRHSS103116 (G06) |
| CCCGUAAUUAUGUGGUGACAGAUCA | Invitrogen | EGFRHSS103114 (G09) |
| CCN1 F- GGTCAAAGTTACCGGGCAGT R- GGAGGCATCGAATCCCAGC | In house | n/a |
| DUSP1 F- GCCTTGCTTACCTTATGAGGAC R-GGGAGAGATGATGCTTCGCC | In house | n/a |
| FOS F- AGGAGGGGAGCTGACTGATACACT R- TTTCCTTCTCCTTCAGCAGGTT | In house | n/a |
| JUNB F- ACGACTCATACACAGCTACGG R- GCTCGGTTTCAGGAGTTTGTAGT | In house | n/a |
| TIMP3 F- CATGTGCAGTACATCCATACGG R- CATCATAGACGCGACCTGTCA | In house | n/a |
| EGR1 F- GAGAAGGTGCTGGTGGAGAC R- CACAAGGTGTTGCCACTGTT | In house | n/a |
| BCL10 F- GTGAAGAAGGACGCCTTAGAAA R- TCAACAAGGGTGTCCAGACCT | In house | n/a |
| CTGF F- CAGCATGGACGTTCGTCTG R- AACCACGGTTTGGTCCTTGG | In house | n/a |
| MCL1 F-ATCTCTCGGTACCTTCGGGAGC R- GCTGAAAACATGGATCATCACTCG | In house | n/a |
| DUSP6 F- CCGCAGGAGCTATACGAGTC R- CGTAGAGCACCACTGTGTCG | In house | n/a |
| ABHD5 F- GCTGCTGCTTACTCGCTGAA R- TCTGATCCAAACTGGAATTGGTC | In house | n/a |
| KDM6B F- CACCCCAGCAAACCATATTATGC R- CACACAGCCATGCAGGGATT | In house | n/a |
| MXD1 F- CGTGGAGAGCACGGACTATC R- CCAAGACACGCCTTGTGACT | In house | n/a |
| NDRG1 F CTCCTGCAAGAGTTTGATGTCC - R- TCATGCCGATGTCATGGTAGG | In house | n/a |
| SPRY2 F- CCTACTGTCGTCCCAAGACCT R- GGGGCTCGTGCAGAAGAAT | In house | n/a |
| ID4 F- TGCCTGCAGTGCGATATGAA R- GCAGGTCCAGGATGTAGTCG | In house | n/a |
| FGFR2b F- AACGGGAAGGAGTTTAAGCAG R- CTCGGTCACATTGAACAGAG | In house | n/a |
| BETA ACTIN F- TGGAACGGTGAAGGTGACAG R- AACAACGCATCTCATATTTGGAA | In house | n/a |
| GAPDH F- CAATGACCCCTTCATTGACC R- GACAAGCTTCCCGTTCTCAG | In house | n/a |

**Reagents and Tools table**   (continued)

| Reagent or Resource | Source | Identifier |
|---|---|---|
| Recombinant DNA | | |
| EGFR (pRK5-EGFR) | Addgene | Plasmid #65225 |
| EGFRT693A | Mutagenesis of above | |
| eGFP-Rab11 | Addgene | Plasmid #12674 |
| eGFP-Rab11_S52N | Mutagenesis of above | |
| Dynamin_K44a-eGFP | Mutagenesis of Addgene plasmid | Plasmid # 34680 |
| HA-FGFR1c | Francavilla *et al* (2009) | PI: Dr Cavallaro |
| HA-FGFR2b | Francavilla *et al* (2013) | PI: Prof Olsen |
| HA_FGFR2b_Y656F/Y657F | Francavilla *et al* (2013) | PI: Prof Olsen |
| HA-FGFR4 cloned using human cDNA with primers F-GGGGCCCAGCCGGCCAGACTGGAGGCCTCTGAGGAAGTGGAGCTTGAGCC R -GTCGACCTGCAGTGTCTGCACCCCAGACCCGAAGGGGAAGGAGCTGGATCC | Generated for this study | n/a |
| Software and Algorithms | | |
| Fiji- Image J version: 1.52p | Schindelin *et al* (2012) | https://imagej.net/Fiji |
| GraphPad Prism version 8.0.0 | GraphPad Software | www.graphpad.com |
| MaxQuant version 1.5.6.5 | Cox and Mann (2008) | http://www.coxdocs.org/doku.php?id=maxquant:start |
| WebGestalt 2019 | Liao *et al* (2019) | http://www.webgestalt.org/ |
| Perseus versions 1.6.5.0 or 1.6.2.1.: | Tyanova *et al* (2016) | http://www.coxdocs.org/doku.php?id=maxquant:start |
| Cytoscape version 3.7.2 | Shannon *et al* (2003) | https://www.cytoscape.org |
| STRING version 11 | Szklarczyk *et al* (2019) | https://string-db.org/ |
| R framework | R Core Team (2018) | https://www.r-project.org/ |
| Other | | |
| Confocal Microscope Leica Sp8 Inverted | Lecia | |
| Mx3000P qPCR machine | Agilent | |
| UltiMate® 3000 Rapid Separation LC | Dionex | |
| QE-HF LC-MS/MS | Thermo Fisher Scientific | |

## Methods and Protocols

### Experimental models

#### Cell culture and SILAC labelling

Human breast cancer cell lines were purchased from ATCC, authenticated through short tandem repeat (STA) analysis of 21 markers by Eurofins Genomics, checked monthly for mycoplasma via a PCR-based detection assay (Venor®GeM—Cambio) and grown in the indicated media supplemented with 2 mM L-glutamine and 100 U/ml penicillin, 100 μg/ml streptomycin, and 10% foetal bovine serum. MCF-7 was grown in DMEM/F12. MDA-MB-415 and BT20 were grown in DMEM. HCC1937, T47D and BT549 were grown in RPMI. 1 mM sodium pyruvate was added to T47D.

For quantitative mass spectrometry, BT549 or T47D cells were labelled in SILAC RPMI (PAA Laboratories GmbH, Germany) supplemented with 10% dialyzed foetal bovine serum (Sigma), 2 mM glutamine (Gibco), 100 U/ml penicillin and 100 μg/ml streptomycin for 15 days to ensure complete incorporation of amino acids, which was verified by MS analysis. Three cell populations were obtained: one labelled with natural variants of the amino acids (light label; Lys0, Arg0), the second one with medium variants of the

amino acids (medium label; L-[13C6] Arg (+6) and L-[2H4]Lys (+4); Lys4/Arg6) and the third one with heavy variants of the amino acids (heavy label; L-[13C6,15N4]Arg (+10) and L-[13C6,15N2]Lys (+8); Lys8/Arg10). The light amino acids were from Sigma, whilst their medium and heavy variants were from Cambridge Isotope Labs (Massachusetts, US).

#### Breast cancer organoid culture and protein isolation

Organoids were generated from a triple-negative breast cancer PDX tumour, BB6RC37 (Eyre *et al*, 2016). The tumours were minced and digested using a tumour dissociation kit (Miltenyi Biotec) on an orbital shaker at 37°C for 1–2 h. The cells were sequentially strained through 100-μm and 40-μm meshes. 50,000 cells were resuspended in 50 μl cold growth factor-reduced Matrigel (Corning 354230), set as domes in a 24-well plate for 30 min and cultured at 37°C in media as defined by (Sachs *et al*, 2018). The organoids were cultured in media with or without FGF7/10 for 14 days, and EGF/Heregulin were removed from the media 24 h before lysates were obtained. Lysates were prepared by mechanically disaggregating the domes and digesting the Matrigel for 1 h using dispase at 37°C (Stem Cell Technologies, 7913). Cells were washed in PBS and

resuspended in lysis buffer as previously described (Santiago-Gomez *et al*, 2019).

### Quantitative phosphoproteomics

#### Experimental design and sample preparation

*TPA1*: for each cell line and each stimulus, we analysed duplicates for each time point, considering both 1- and 8-min. time points as representative of early signalling. Therefore, we compared four label-free samples for each stimulus in each cell line (Datasets EV1 and EV2). The cell pellet was dissolved in denaturation buffer (6 M urea, 2 M thiourea in 10 mM HEPES pH 8). We obtained 1 mg of proteins from each sample. Cysteines were reduced with 1 mM dithiothreitol (DTT) and alkylated with 5.5 mM chloroacetamide (CAA). Proteins were digested with endoproteinase Lys-C (Wako, Osaka, Japan) and sequencing grade modified trypsin (modified sequencing grade, Sigma) followed by quenching with 1% trifluoroacetic acid (TFA). Peptides were purified using reversed-phase Sep-Pak C18 cartridges (Waters, Milford, MA) and eluted with 50% acetonitrile (ACN). After removing ACN by vacuum concentrator at 60°C, peptides were suspended in phosphopeptide immunoprecipitation buffer (50 mM MOPS pH 7.2, 10 mM sodium phosphate, 50 mM NaCl) and dissolved overnight. Clarified peptides were transferred in a new tube containing immobilized phosphorylated tyrosine antibody beads (pY100-AC, Cell Signalling Technologies) and incubated for two hours at 4°C. After five washes with immunoprecipitation buffer followed by two washes with 50 mM NaCl, the enriched peptides were eluted from the beads three times with 50 μL of 0.1% TFA, loaded on C18 STAGE-tips, and eluted from STAGE-tips with 20 μL of 40% ACN followed by 10 μL 60% ACN and reduced to 5 μL by SpeedVac and 5 μL 0.1% formic acid (FA) 5% ACN added. Peptides from the supernatant were purified using reversed-phase Sep-Pak C18 cartridges (Waters, Milford, MA) and eluted with 50% ACN and further enriched for phosphorylated serine- and phosphorylated threonine-containing peptides, with Titansphere chromatography. Six mL of 12% TFA in ACN was added to the eluted peptides and subsequently enriched with TiO2 beads (5 μm, GL Sciences Inc., Tokyo, Japan). The beads were suspended in 20 mg/mL 2,5-dihydroxybenzoic acid (DHB), 80% ACN, and 6% TFA and the samples were incubated in a sample to bead ratio of 1:2 (w/w) in batch mode for 15 min with rotation. After 5-min centrifugation, the supernatant were collected and incubated a second time with a twofold dilution of the previous bead suspension. The beads were washed with 10% ACN, 6% TFA followed by 40% ACN, 6% TFA and collected on C8 STAGE-tips and finally washed by 80% ACN, 6% TFA. Elution of phosphorylated peptides was done with 20ul 5% NH3 followed by 20 μL 10% NH3 in 25% ACN, which were evaporated to a final volume of 5 μL in a sped vacuum. The concentrated phosphorylated peptides were acidified with addition of 20 μL 0.1% TFA, 5% ACN and loaded on C18 STAGE-tips. Peptides were eluted from STAGE-tips with 20 μL of 40% ACN followed by 10 μL 60% ACN and ACN and reduced to 5 μL by SpeedVac and 5 μL 0.1% FA, 5% ACN added.

A small amount of the eluted peptides (1%) was taken for proteome analysis before enrichment of phosphorylated peptides: after evaporation in a speed vacuum, 40 μl of 0.1% TFA, 5% ACN were added followed by MS analysis.

*TPA2*: we analysed label-free triplicates for each condition, T47D depleted or not of TTP or RCP and stimulated or not with FGF10. Cells were washed with PBS and lysed at 4°C in ice-cold 1% triton lysis buffer supplemented with Pierce protease inhibitor tablet (Life Technologies) and phosphatase inhibitors: 5 nM Na3VO4, 5 mM NaF and 5 mM β-glycerophosphate. Proteins were precipitated overnight at −20°C in fourfold excess of ice-cold acetone. The acetone-precipitated proteins were solubilized in denaturation buffer (10 mM HEPES, pH 8.0,6 M urea, 2 M thiourea), and 5 mg of proteins was reduced, alkylated and digested, as described above. All the steps were performed at room temperature. The peptide mixture was desalted and concentrated on a C18-Sep-Pak cartridge, eluted and enriched with TiO2 beads, as described above.

*TPA3*: we analysed duplicates of SILAC-labelled BT549, transfected and treated as described in Fig 2A. We followed the same procedure described for TPA2 with the only difference that 5 mg of each SILAC-labelled lysates was mixed in equal amount before digestion and TiO2 chromatography.

*EGFR- and EGFR_T693A-expressing T47D cells*: we analysed duplicates of SILAC-labelled T47D transfected and treated as described in Fig 6A. We followed the same procedure described for TPA1 with the only difference that 5 mg of each SILAC-labelled lysates was mixed in equal amounts before digestion and phosphorylated tyrosine enrichment followed by TiO2 chromatography and peptides purification.

### Mass spectrometry

Purified peptides were analysed by LC-MS/MS using an UltiMate® 3000 Rapid Separation LC (RSLC, Dionex Corporation, Sunnyvale, CA) coupled to a QE-HF (Thermo Fisher Scientific, Waltham, MA) mass spectrometer. Mobile phase A was 0.1% FA in water, and mobile phase B was 0.1% FA in ACN and the column was a 75 mm x 250 μm inner diameter 1.7 μM CSH C18, analytical column (Waters). A 1 μl aliquot of the sample (for proteome analysis) or a 3 μl aliquot was transferred to a 5 μl loop and loaded on to the column at a flow of 300 nl/min at 5% B for 5 and 13 min, respectively. The loop was then taken out of line and the flow was reduced from 300 nl/min to 200nl/min in 1 min., and to 7% B. Peptides were separated using a gradient that went from 7% to 18% B in 64 min., then from 18% to 27% B in 8 min. and finally from 27% B to 60% B in 1 min. The column was washed at 60% B for 3 min. and then re-equilibrated for a further 6.5 min. At 85 min, the flow was increased to 300nl/min until the end of the run at 90min. Mass spectrometry data were acquired in a data directed manner for 90 min in positive mode. Peptides were selected for fragmentation automatically by data-dependent analysis on a basis of the top 8 (phosphoproteome analysis) or top 12 (proteome analysis) with m/z between 300 and 1750Th and a charge state of 2, 3 or 4 with a dynamic exclusion set at 15 s. The MS resolution was set at 120,000 with an AGC target of 3e6 and a maximum fill time set at 20ms. The MS2 resolution was set to 60,000, with an AGC target of 2e5, and a maximum fill time of 110 ms for Top12 methods, and 30,000, with an AGC target of 2e5, and a maximum fill time of 45 ms for Top8 analysis. The isolation window was of 1.3Th (2.6 Th for SILAC-labelled samples), and the collision energy was of 28.

### Raw files analysis

Raw data were analysed by the MaxQuant software suite (Cox & Mann, 2008) (https://www.maxquant.org; version 1.5.6.5) using the integrated Andromeda search engine (Cox *et al*, 2011). Proteins were identified by searching the HCD-MS/MS peak lists against a target/decoy version of the human UniProt Knowledgebase database that consisted of the complete proteome sets and isoforms (v.2016; https://uniprot.org/proteomes/UP000005640_9606) supplemented with commonly observed contaminants such as porcine trypsin and bovine serum proteins. Tandem mass spectra were initially matched with a mass tolerance of 7 ppm on precursor masses and 0.02 Da or 20 ppm for fragment ions. Cysteine carbamidomethylation was searched as a fixed modification. Protein N-acetylation, N-pyroglutamine, oxidized methionine and phosphorylation of serine, threonine and tyrosine were searched as variable modifications. Protein N-acetylation, oxidized methionine and deamidation of asparagine and glutamine were searched as variable modifications for the proteome experiments. For the quantification of SILAC-labelled samples, labelled lysine and arginine were specified as fixed or variable modification, depending on prior knowledge about the parent ion (MaxQuant SILAC triplet identification). In all the other experiments, label-free parameters were used as described (Cox *et al*, 2014). False discovery rate was set to 0.01 for peptides, proteins and modification sites. Minimal peptide length was six amino acids. Site localization probabilities were calculated by MaxQuant using the PTM scoring algorithm (Olsen *et al*, 2006). The datasets were filtered by posterior error probability to achieve a false discovery rate below 1% for peptides, proteins and modification sites. Only peptides with Andromeda score > 40 were included.

### Data and statistical analysis

All statistical and bioinformatics analyses were done using the freely available software Perseus, version 1.6.5.0 or 1.6.2.1. (Tyanova & Cox, 2018), R framework (R Core Team, 2018), Bioconductor R-package LIMMA (Bolstad *et al*, 2003), WebGestalt (Liao *et al*, 2019), STRING (Szklarczyk *et al*, 2019), Cytoscape (version 3.7.2) (Shannon *et al*, 2003). All measured peptide intensities were normalized using the "normalizeQuantiles" function from the Bioconductor R-package LIMMA, which normalizes the peptide intensities such that each quantile for each sample is set to the mean of that quantile across the dataset, resulting in peptide intensity distributions that are empirically identical. Each dataset was normalized individually. Subsequent data analysis was performed using Microsoft Office Excel, R and Perseus. For the SILAC datasets, we used the normalized SILAC ratios from MaxQuant output txt files. Only peptides with localization probabilities higher than 0.75 (class I, shown in Datasets EV1, EV3–EV6; Olsen *et al*, 2006) were included in the downstream bioinformatics analysis. Pearson correlation was calculated in R. For TPA1, we impute missing values using Perseus default settings, we subtracted the control from log intensity values in order to be able to compare all the cell lines against each other and we used the median for each condition. Hierarchical clustering based on correlation was performed after multi-sample ANOVA test with default parameters in Perseus. For TPA2, we calculated the median and then considered only rows with four valid values, followed by hierarchical clustering based on Euclidean distance in Perseus. For TPA3 and the EGFR/EGFR_T693A T47D dataset, we imputed missing values using Perseus default settings and then calculated the median, followed by hierarchical clustering based on Euclidean distance in Perseus. Clusters used in the follow-up analysis were defined by Perseus and manually checked.

The enrichment of KEGG or GO terms was performed in WebGestalt using the ORA default parameters, and significantly overrepresented terms within the data were represented in bar plots. The relation of genes to other diseases was based on the database DISEASES (Pletscher-Frankild *et al*, 2015).

All the protein interaction networks were obtained using the STRING protein interaction database using high confidence, and interactions derived from the Experiments and Databases evidence channels. Data visualization was performed using the software Cytoscape. The Venn diagram was created using the web tool http://bioinformatics.psb.ugent.be/cgi-bin/liste/Venn/calculate_venn.htpl.

## Biochemical assays

### RNA isolation and real-time PCR analysis

RNA from cell lines was isolated with TRIZOL® (Invitrogen). After chloroform extraction and centrifugation, 5 μg RNA was DNase treated using RNase-Free DNase Set (Qiagen) and 1 μg of DNase treated RNA was then taken for cDNA synthesis using the Protoscript I first strand cDNA synthesis kit (New England Biolabs). Selected genes were amplified by quantitative real-time PCR (RT–qPCR) using Sygreen (PCR Biosystems). Relative expression was calculated using the delta-delta CT methodology, and beta-actin was used as reference housekeeping gene. Sequences for primers used can be found in the accompanying Reagent Table. qPCR machine used was Applied Biosystems MX300P.

### Transfection and RNA interference

All transfections were carried out in Gibco opti-MEM glutamax reduced serum media (Thermo Fisher Scientific). For RNA interference, all cells were transfected using Lipofectamine RNAiMax (Thermo Fisher Scientific), according to manufacturer instructions. Validated double-stranded stealth siRNA oligonucleotides were used for RNA interference. siRNA Universal Negative Control #2 (Sigma-Aldrich) was used as a control in all RNA interference experiments. BT549 and BT20 cells were transfected using Lipofectamine 3000 (Thermo Fisher Scientific) according to the manufacturer's instructions, 24 h after RNA interference transfection where indicated. T47D cells were transfected using Escort IV according to manufacturer instructions, same as above. Assays were performed 36 h after transfection, as previously described (Francavilla *et al*, 2016). Where assays were performed more than 36 h after transfection, RNAi and expression were assessed at time of assay to confirm expression.

### Cell lysis, protein immunoprecipitation and western blotting

Cells were serum starved overnight in serum-free medium and stimulated for the indicated time points with 100 ng/ml of FGF7, FGF10, EGF or TGFα. Ligands were replenished every 24 h for long-term (24–72 h) stimulation. Where indicated, cells were pre-incubated for 2 h with 100 nM PD173074, 500 nM AG1478, 20 μM U1206, 1 μM MEK162 or 10 μM BMS582949. Control cells were preincubated with DMSO alone. After stimulation, cell extraction and immunoblotting were performed as previously described (Francavilla *et al*, 2016). Proteins were resolved by SDS–PAGE and transferred to nitrocellulose membranes (Protran, Biosciences). Proteins of interest were visualized using specific antibodies, followed by

peroxidase-conjugated secondary antibodies and by an enhanced chemiluminescence kit (Amersham Biosciences). Blots were visualized either using film exposure or the Universal Hood II Gel Molecular Imaging System (Bio-Rad). Each experiment was repeated at least three times and produced similar results.

Immunoprecipitation of FGFR2 from cell extracts was performed as previously described (Francavilla *et al*, 2016), using anti BEK (Santa Cruz Biotechnology, sc-121). Each experiment was repeated at least three times and produced similar results.

### Biotinylation assays

Biotinylation pull down experiments were performed as described previously (Lobingier *et al*, 2017). Briefly biotinylation experiments were performed by transfecting GFP-Rab11-APEX2 constructs in to 2 million cells plated in 10-cm dishes. Cells were pre-incubated (40 min) with biotin phenol (Iris Biotech), after stimulation with ligands, hydrogen peroxide (Sigma-Aldrich) was added for 1 min before quenching with Trolox (Sigma-Aldrich) and sodium ascorbate (VWR) during ice-cold lysis. A 2-hour RT pull down with streptavidin beads was then performed running the supernatant against the bound proteins.

### Proliferation assays

#### Incucyte cell proliferation assay

Indicated cell lines were seeded into 24-well plates at a density of 15,000–20,000 cells per well, depending on growth rate and the design of the experiment. After plating cells were starved and stimulated with indicated ligands every 24 h and imaged every hour using the Incucyte ZOOM (Essen Bioscience), phase-contrast images were analysed to detect cell proliferation based on cell confluence. And average confluency value over 4 h was used to determine the starting confluency from which a relative growth change was calculated. Statistical analysis was performed at the endpoint across repeats, as indicated in the Fig legends.

#### Crystal Violet

Indicated cells were stained after experimentation by being fixed with 0.5% w/v crystal violet (Sigma) in 4% w/v paraformaldehyde/PBS for 30 min. Fixed cells were then solubilized in 2% w/v SDS/PBS and absorbance measured at 595 nm using Synergy H1 microplate reader (BioTek). Statistical analysis was performed at the endpoint across repeats, as indicated in the Fig legends.

#### EdU incorporation

Indicated cells were labelled with 20 µM 5-ethynyl-2'-deoxyuridine (EdU) for 4 h and processed following the manufacturer's protocol (Click-iT® EdU Alexa Fluor® 488 Imaging Kit, Thermo Fisher). Prior to imaging, cells were then stained with 5 ng/ml Hoecsht 3342 for 15 min. Stained cells were analysed using a using a Leica microscope system. Statistical analysis was performed at the endpoint across repeats, as indicated in the Fig legends.

### Invasion assay

Rat tail-derived collagen I (Corning) was supplemented with 25 µg/ml human fibronectin (Sigma) in DMEM and polymerized in 8-µm Transwell inserts (Corning) for 30 min at room temperature followed by 30 min at 37°C/5% CO2. 5 x 10⁴ BT20 cells were seeded on the reverse of each insert and incubated for 6 h at 37°C/5% CO2. Inserts were gently washed and placed in serum-free DMEM and the upper chamber filled with DMEM supplemented with 10% FCS (Life Technologies) and either PBS or 100 ng/ml FGF10 (PeproTech). After 72 h, cells were stained with 500 ng/ml Calcein AM (Thermo Fisher) for 1 h and visualized by Leica Sp8 inverted confocal microscopy in serial sections of 20 µm. Fluorescence intensity of each section was determined using ImageJ v. 1.52p (Schindelin *et al*, 2012) and proportion of invading cells estimated by comparing the total intensity beyond 40 µm with the total overall intensity per insert using GraphPad PRISM version 8.0.0. Statistical analysis was performed at the endpoint across repeats, as indicated in the Fig legends.

### Immunofluorescence

Immunofluorescence staining was performed as previously described (Francavilla *et al*, 2016). To detect HA-FGFR2b or endogenous FGFR2, we incubated cells with 10 µg/ml of anti-HA (Covance) or anti-FGFR2 antibody (Cell Signalling) for 45 minutes with gentle agitation. The binding of the antibody did not activate receptor signalling in untreated cells nor induced receptor internalization (see control cells in Fig 1), as previously reported (Francavilla *et al*, 2009). After stimulation, cells were incubated at 37°C for different time points. When indicated, each inhibitor was added prior stimulation. At each time point, non-permeabilized cells were either fixed to visualize the receptor on the cell surface (plasma membrane) or acid-washed in ice-cold buffer (50 mM glycine, pH 2.5) to remove surface-bound antibody. Acid-washed cells were then fixed and permeabilized to visualize the internalized receptor (cytoplasm). Finally, to detect FGFR2b cells were stained with AlexaFluor488-conjugated donkey anti-mouse or anti-rabbit (Jackson ImmunoResearch Laboratories). Nuclei were stained with DAPI. Coverslips were then mounted in mounting medium (Vectashield; Vector Laboratories).

For co-localization experiments, cells were acid-washed, fixed, permeabilized with 0.02% saponin (Sigma), treated with a primary antibody against FGFR2, EGFR, TTP, RCP, phosphorylated T693 EGFR, EEA1 for 60 min at 37 °C and stained with AlexaFluor488 (or 568 or 647)-conjugated donkey anti-mouse or anti-rabbit. Samples either expressing GFP-tagged proteins or treated with TRITC-transferrin or Alexa 647-transferrin (to stain transferrin receptor, Tf-R), added to the medium at a final concentration of 50 µg/mL, were kept in the dark. Nuclei were stained with DAPI. Coverslips were then mounted in mounting medium (Vectashield; Vector Laboratories).

All the images were acquired at room temperature on a Leica TCS SP8 AOBS inverted confocal using a 100x oil immersion objective and 2.5x confocal zoom. The confocal settings were as follows: pinhole, 1 airy unit, format, 1,024 × 1,024. Images were collected using the following detection mirror settings: FITC 494-530nm; Texas red 602-665nm; Cy5 640-690nm. The images were collected sequentially. Raw images were exported as.lsm files, and adjustments in image contrast and brightness were applied identical for all images in a given experiment using the freely available software ImageJ v. 1.52p (Schindelin *et al*, 2012).

### Quantification of the recycling assay

Quantification of recycling was performed as described (Francavilla *et al*, 2016). For each time point and each treatment, the presence (total) and the localization (cell surface versus internalized) of HA-FGFR2 or endogenous FGFR2 were assessed in at least seven randomly chosen fields. Approximately 100 cells per condition (both acid-washed and not) were analysed from three independent

experiments. The results are expressed as the percentage of receptor-positive cells (green) over total cells (corresponding to DAPI-stained nuclei) and referred to the values obtained at time zero. Statistical analysis was performed across repeats, as indicated in the Fig legends.

*Quantification of expression fraction, overlap fraction and co-localization*

Images were pre-processed using an "À trous" wavelet band-pass filter to reduce the contribution of high-frequency speckled noise to the co-localization calculations. Pixel intensities were then normalized from the original 8-bit range [0,255] to [0,1]. To ensure that co-localization was only computed in well-determined regions of interest (ROI), we used the Fiji/ImageJ (Schindelin *et al,* 2012) built-in ROI manager to create and record these regions.

To measure differences in expression over time or between conditions, we computed the fractions of expressed red marker (R), green marker G. or far-red marker F. pixels over a region of interest ($N_R$, $N_G$ or $N_F$ pixels with a strictly positive intensity in $N$ pixels):

$$F_R = \frac{N_R}{N} \quad F_G = \frac{N_G}{N} \quad F_F = \frac{N_F}{N}$$

To quantify the overlap fraction between two (R and G) or three (R, F and G) markers, we first multiplied the (normalized) channel intensities together, i.e $I_{RG} = I_R \times I_G$ and $I_{RFG} = I_R \times I_F \times I_G$ to compute a new image whose intensity increases to 1 where the markers strongly overlap and decreases or becomes null for non-overlapping pixels. Our overlap fraction coefficient (OF) becomes the fraction of strictly positive pixels in the combined image over the number of pixels in the region of interest.

$$OF_{RG} = \frac{N_{RG}}{N} \quad OF_{RF} = \frac{N_{RF}}{N} \quad OF_{RFG} = \frac{N_{RFG}}{N}$$

Finally, to quantify the actual level of co-localization between two markers (e.g. R and G), we used the Manders co-localization coefficients (MCC) M1 and M2 (Manders *et al,* 1996). M1 measures the fraction of the R marker in compartments that also contain the G marker, and M2, the fraction of the G marker in compartments that also contain the R marker. Lower-bound thresholds for pixel intensities $I_R$ and $I_G$ were automatically determined using the Costes method (Costes *et al,* 2004).

Briefly,

$$M1 \cong \frac{\sum_{I_R > T_R} I_{R,C}}{\sum_{All\ I_R} I_R} \quad M2 \cong \frac{\sum_{I_C > T_C} I_{G,C}}{\sum_{All\ I_C} I_G}$$

where with $T_R$ and $T_G$ are the threshold set by the automated Costes algorithm for the R and G channels, and $I_{R,C}$ and $I_{G,C}$ pixels are co-localized if their intensity in the reciprocal channel is above $T_R$ or $T_G$ set for that channel.

To measure the simultaneous overlap of our three, red, far-red and green markers (R, F, G), we first used the overlap image between marker R and marker F as defined above (i.e. e. $I_{F,R} = I_F \times I_R$). We then measured the MCC co-localization parameter of this combined image against a Green marker using the MCC formulae above, together with the Costes method to determine the $T_{FR}$ and $T_G$ thresholds.

The scripts for the quantification of co-localization were written in the Python language, and the code for Costes-adjusted MCC was taken verbatim from the CellProfiler (McQuin *et al,* 2018) code base.

Student's t-test was subsequently used to determine the difference in pixel overlap fraction or Manders (Costes) coefficient between different experimental conditions in Figs 1 and Figs 5, and Appendix Fig S4 and S6.

## Data availability

The mass spectrometry proteomics data in Thermo Scientific's *.raw format have been deposited to the ProteomeXchange Consortium via the PRIDE (Perez-Riverol *et al,* 2019) partner repository with the dataset identifier PXD018184. Submission details: Project Name: Proximal Phosphoproteomics Approaches revealed a FGFR-EGFR functional cross-talk Project accession: PXD018184. To download: https://www.ebi.ac.uk/pride/archive/projects/PXD018184. The scripts for the quantification of overlap fraction and co-localization are available on Github at the following address: https://github.com/manbio/smith_ferguson_coloc.

Expanded View for this article is available online.

## Acknowledgements

We thank Profs. Lowe and Woodman, and Drs Caswell, Tournier and Lopez-Castejon for reading the manuscript and the Bioimaging and the Bio-MS Facilities (University of Manchester). Research in CF lab is supported by Wellcome Trust (WT Sir Henry Dale fellowship 107636/Z/15/Z), the Biotechnology and Biological Sciences Research Council (BB/R015864/1), and Medical Research Council (MR/T016043/1). PhD students are supported by BBSRC Doctoral Training Programme (HF and JW: BB/M011208/1); Wellcome Trust (CB: 210002/Z/17/Z); CR-UK Non-Clinical Training Award (JP: A27445); and NIHR Manchester Biomedical Research Centre non-clinical PhD Studentship (TK: IS-BRC-1215-20007). RBC is supported by Cancer Research UK and Breast Cancer Now (MAN-Q2-Y4/5).

## Author contributions

MPS performed experiments, supervised HRF and helped writing the manuscript. HRF, JF, KMK, TK, CB, JP and SC performed experiments. EZ quantified imaging data. JW helped with data analysis. PF contributed to sample preparation for MS analysis. SW and DK performed the MS experiments and provided technical advice. RBC supervised TK. CF conceptualized the study, performed experiments, analysed the data, supervised PhD students and wrote the manuscript. All the authors contributed to writing and approving the manuscript.

## Conflict of interest

The authors declare that they have no conflict of interest.

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
