## [Review Process File · The EMBO Journal]

Reciprocal Priming between Receptor Tyrosine Kinases at Recycling Endosomes Orchestrates Cellular Signalling Outputs

Michael Smith, Harriet Ferguson, Jennifer Ferguson, Egor Zindy, Katarzyna Kowalczyk, Thomas Kedward, Christian Bates, Joseph Parsons, Joanne Watson, Sarah Chandler, Paul Fullwood, Stacey Warwood, David Knight, Robert Clarke, and Chiara Francavilla

DOI: [10.15252/embj.2020107182](https://doi.org/10.15252/embj.2020107182)

Corresponding author(s): Chiara Francavilla (chiara.francavilla@manchester.ac.uk)

Review Timeline:

Submission Date:	29th Oct 20
Editorial Decision:	8th Dec 20
Revision Received:	20th Mar 21
Editorial Decision:	20th Apr 21
Revision Received:	27th Apr 21
Accepted:	28th Apr 21

Editor: Hartmut Vodermaier

Transaction Report:

Thank you for submitting your manuscript on reciprocal RTK priming for our consideration. I have now heard back from three expert referees who had agreed to review it. As you will see from the comments copied below, all three referees appreciate the interest and overall quality of the study. We shall therefore be happy to consider this work further for EMBO Journal publication, following adequate revision of a number of points raised, in particular by referees 2 and 3. As their main issues are well-explained in the below reports, I will not go through them in detail here, but would of course be happy to discuss any questions you may have related to the reviewers' reports or your revision plans further with you via email and/or direct call. In light of the still unpredictable research restrictions around the COVID-19 pandemic situation, we might also discuss possible extensions of the default three-months revision deadline. Our 'scooping protection' (meaning that competing work appearing elsewhere in the meantime will not affect our considerations of your study) would of course remain valid also during an eventually extended revision.

Referee #1:

Francavilla and co-workers use advanced phosphoproteomics technology to investigate cross talk between RTK pathways. In contrast to previous studies they incorporate a novel spatial aspect, which they call TPAs for Trafficking Phosphoproteomics Approaches. Their system are breast cancer cell lines where they focus on signaling events in recycling endosomes. Specifically they focus on the two cell lines T47D and BT20 as they represent different breast cancer sub-types. Cells were treated with different FGFR ligands and for different time durations and deep phosphoproteomes were generated. Systems analysis of their data led them to discover that threonine 693 of EGFR correlates uniquely with RTK recycling suggesting a functional cross talk

between FGF and EGF signaling pathways. This finding is the key one that the authors focus on. They verify it in different systems, including an organoid one and use a variety of functional follow ups (siRNA, inhibitors) to establish a crucial role for this phospho site. The authors drill down on the mechanism in a number of ways, most importantly with and EGFR point mutant for threonine 693. They build a 'reciprocal priming' model that summarizes the interaction of FGFR and EGFR via threonine 693.

I find the study very elegant and done at a very high level of expertise in all the different experimental aspects that are involved. Not only did the authors perform cutting edge proteomics and phospho-proteomics but they also did so in many different settings and perturbations. Far from leaving it at the resource stage, the authors discovered a key functional site, which they generalize successfully into their 'reciprocal priming' model, which seems to be generalizable.

Minor points:

- although common in the field, the authors could consider using less jargon, i.e. abbreviations that the reader has to remember but many won't.
- given the central importance of this EGFR site (T693), is it evolutionary conserved and how far?
- Can the authors speculate on structural consequences of this phosphorylation and how it might explain the cross talk?
- 'Results' appears twice and there are no figure numbers.

Referee #2:

In this study the authors have used quantitative phosphoproteomics to analyse the signalling from the FGF receptor tyrosine kinase that is regulated by endocytic recycling. They have discovered that FGFR ligands that promote FGFR recycling induce phosphorylation of another receptor tyrosine kinase, the EGFR, on Threonine 693 and this phosphorylation modulates both EGFR and FGFR trafficking and signalling outputs. These findings are novel and intriguing and may have relevance to breast cancer, where targeting a single tyrosine kinase has limited efficacy. Understanding the interplay between different receptor kinases and how they are regulated by endocytic recycling may lead to the exploration of novel therapeutic avenues. Additionally, this study provides valuable data sets that will be of interest to multiple researchers in the fields of receptor tyrosine kinase signalling and trafficking.

Overall the study is well presented, the data convincing and represents an impressive mix of proteomics, microscopy and biochemical analysis. I am convinced that phosphorylation of EGFR on T693 requires FGFR recycling and that this phosphorylation regulates both FGFR and EGFR trafficking and signalling. Where I am less convinced is that the EGFR phosphorylation occurs in recycling endosomes:

It is clear that FGF10-stimulated FGFR2 recycling is required for phosphorylation of EGFR T693 (FGF7, which doesn't promote FGFR2 recycling doesn't induce T693 phosphorylation, and T693 phosphorylation is inhibited by TTP or RCP depletion or expression of dominant negative Rab11, all of which inhibit FGF10-mediated FGFR2 recycling). Erk activation is necessary for T693 phosphorylation (as it is prevented by Erk inhibitors) but is not sufficient because FGF7 promotes Erk activation without promoting EGFR T693 phosphorylation. Notably, FGF10-mediated Erk activation appears to require recycling as it is inhibited by depletion of TTP and RCP. For me this latter result is extremely intriguing. Ideally I would like to see as an additional control the effects of dominant negative dynamin expression on FGF10-mediated Erk activation. When recycling is inhibited with TTP or

RCP there appears in some cases to be some degradation of FGFR which could contribute to inefficient Erk activation so it would be nice to check that inhibition of endocytosis (when there should be no loss of FGFR) also blocks FGF10-mediated Erk activation. If the authors have already done this it should be shown. The authors suggest that EGFR T693 phosphorylation occurs in recycling endosomes, as evidenced by anti-phT693 staining in endosomes by IF in FGF10 stimulated cells. I think it is also possible that phosphorylation of EGFR on T693 occurs at the plasma membrane leading to EGFR endocytosis and accumulation in recycling endosomes. I think that the data is consistent with this possibility which is acknowledged in the discussion, but the summary states: "We discovered reciprocal priming between FGFRs and Epidermal Growth Factor Receptor (EGFR) within recycling endosomes". Dominant negative dynamin also blocks T693 phosphorylation - this would be expected to accumulate both FGFR and EGFR on the cell surface, supporting the idea that EGFR:PDGFR priming might occur in recycling endosomes but dominant negative dynamin could prevent the Erk activation that is also necessary for T693 phosphorylation.

I would very much like to be convinced that this priming event occurs in endosomes but I am not sure that this has been conclusively shown. Nevertheless, I am convinced that reciprocal priming between EGFR and FGFR is occurring that depends on recycling. I think therefore that the authors should be very careful in their conclusions and, given that there is a lot of data in this paper, a summary table of the conditions under which FGFR recycles, Erk is activated, EGFR is phosphorylated on T693 etc, EGFR co-ips with FGFR might help. A summary figure is presented but I think that there are some assumptions made in that figure, eg has RCP been shown to mediate FGFR interaction with EGFR? Also a direct interaction between EGFR and FGFR on the plasma membrane is implied. Has this been shown (other than by co-ip which does not demonstrate a direct interaction)?

Additional points are detailed below:

Fig. 1H and I: following siRNA-mediated depletion of RCP or TTP, not only is FGF10-stimulated FGFR recycling inhibited but FGFR degradation appears to be promoted. This is potentially important as changes in phosphoproteome could be due to reduced numbers of FGFR in the cell, in addition to reduced accumulation in recycling endosomes. Is this reproducibly the case? If so it should be noted in the text.

I find the graphical presentation of the recycling assays (eg Fig 1I and Fig S3B) hard to read- the data points and error bars are difficult to make out.

Fig. 3C The FGF10 treated cells expressing HA-FGFR2 appear to have far more FGFR2 than the other cells, which could be contributing to the elevated EGFR T693 phosphorylation.

The authors state that "EGFR_T693 phosphorylation is uniquely dependent on MEK-ERK signaling upon FGF10 but not TGF α stimulation." However, it appears from Fig. 3F that EGFR T693 phosphorylation is also substantially reduced following MEK inhibition in TGF α treated cells.

I have some concerns over the Western blot shown in Figure 4B which investigates the effects of pretreatment with FGF for 40 mins on EGFR abundance and signaling. I can see that FGF pretreatment increases the abundance of EGFR, but the effects on Erk phosphorylation appear somewhat marginal and the effect of FGF10 pretreatment appear even more clear on TGF stimulation than EGF stimulation, at variance with the authors assertion that "FGF10 pre-treatment did not affect the response of the EGFR recycling stimulus TGF". Also- FGFR levels appear to be

majorly affected by treatment with EGF and TGF. Additionally, in Figure 4E it appears to me that pretreatment with FGF does increase the level of EGFR following EGF stimulation in cells treated with EGFR inhibitor at variance with the authors statement that "Conversely, when EGFR was stabilized with AG1478 (Gan et al ,2007), the total level of EGFR was not affected by FGF10 pretreatment although ERK activation decreased (Figure 4E). One of the potential issues here is that there are a lot of Western blots shown in this paper with no quantitation. I accept that Western blotting is only semi-quantitative at best, some of the Westerns are confirming data obtained from quantitative phosphoproteomics and some idea of reproducibility can be gleaned from the fact that there is some repetition of key treatments in different figures. Nevertheless, as detailed above, some of the Westerns, particularly in Figure 4, do not seem to agree with what is being described in the text.

The authors state: "Indeed, FGFR2b trafficking is faster in T693A-expressing cells and in cells treated with FGFR and ERK inhibitors, but not EGFR or p38, inhibitors (Figure 5H, Appendix Figure S6H-I)." These figures show that there is reduced intracellular FGFR and increased cell surface receptor at 40 minutes but that this represents faster trafficking is an assumption. For example less FGFR could be endocytosed. Nevertheless, the possibility that phosphorylation of the EGFR delays recycling of FGFR, modifying signaling output is an interesting one. Can the authors speculate on how phosphorylation of EGFR on T693 can affect trafficking of FGFR?

Referee #3:

This is a detailed and creative look into the crosstalk between two receptor tyrosine kinases (RTK). The overarching conclusion of the manuscript is that stimulation of the FGFR with a ligand that induces degradation (FGF10) of the ligand:receptor complex primes the EGFR for enhanced signaling. This study is comprehensive, the data are generally of high quality, and the manuscript is well written. This is a high quality cell signaling study that will be appreciated by the readership of EMBO J.

The manuscript starts with a proteomics approach to look for changes in the phosphorylation profiles of cellular proteins following treatment with ligands that are known to promote receptor recycling (FGF7) versus receptor degradation (FGF10). From those studies, they identify the EGFR as a phospho-substrate following treatment with FGF10. Subsequent biochemical analysis reveals that the EGFR is phosphorylated on Threonine 693 in response to stimulation with FGF10 (degradation ligand). A series of knock down and rescue experiments argues for the need for the EGFR to be phosphorylated on Thr693. Detailed biochemical/pharmacological characterization demonstrates that this phosphorylation is mediated through FGFR2b and utilizes the MEK pathway. Further, phosphorylation of the EGFR on Thr693 slows the rate of ligand stimulated EGFR degradation and sustains its signaling.

In addition, the authors include studies that demonstrate the FGFR2 and EGFR co-localize in early endosomes. In the absence of Thr693 phosphorylation (by expressing a phosphorylation defective mutant receptor: EGFR_T693A), there is reduced EGFR in endosomes. However, the ligand-stimulated FGFR2 and the EGFR can still associate. The authors propose a model in which EGFR phosphorylation at Thr693 dynamically regulates FGFR2b trafficking.

Finally, the authors demonstrate that pretreatment with FGF10 can enhance DNA synthesis and cell invasion in response to an EGFR ligand (TGF α). Both invasion and DNA synthesis require EGFR phosphorylation at Thr693 for this enhanced response.

The major strengths of the manuscript are the multifaceted approaches to testing the hypothesis as well as the use of multiple cell lines. There are only two concerns regarding the data and its interpretation (described below). The biochemical findings are strengthened by the observed changes in cell biology (DNA synthesis, invasion).

There is a concern regarding the micrographs in Figure 5. The colocalization of the receptors are not clear, even with the magnifications that are presented. One suggestion is to show individual channels and the merge of this magnified section. This addition is unlikely to change the interpretation given the quantifications of the co-localization of the fluorophores that accompany the image, but will help the reader visualize the data better.

Second, the kinetics of EGFR phosphorylation and internalization are unclear particularly when the constitutive recycling of the EGFR is considered. An alternative model for these data would be that the FGFR2b phosphorylates the EGFR to prevent its recycling to the plasma membrane, rather than promoting its internalization. Thus, the change in EGFR distribution is not due to an increase in receptor internalization, but from a slowed return to the plasma membrane. The kinetics would be more consistent with this model, but there may be additional evidence that refutes this model. If appropriate, this idea should be incorporated into the discussion.

Minor concerns

In Figure 1H the nuclei are quite variable in size. For instance, the cells treated with FGF10 for 40 minutes have nuclei that are 3-4 times the size of the cells treated with FGF10 for 120 minutes. Is there a scientific reason or technical reason? Are there more representative images?

The figure legends for Fig 5H do not explain how the quantifications were obtained. Are these derived from the micrographs? How many cells and experiments do they represent.

Point-by-point responses to reviewer's comments.

Our responses to reviewer comments are provided below in **blue font** after each comment and changes to the manuscript are visualized by *italic blue font*.

Referee #1:

Francavilla and co-workers use advanced phosphoproteomics technology to investigate cross talk between RTK pathways. In contrast to previous studies they incorporate a novel spatial aspect, which they call TPAs for Trafficking Phosphoproteomics Approaches. Their system are breast cancer cell lines where they focus on signaling events in recycling endosomes. Specifically, they focus on the two cell lines T47D and BT20 as they represent different breast cancer sub-types.

Cells were treated with different FGFR ligands and for different time durations and deep phosphoproteomes were generated. Systems analysis of their data led them to discover that threonine 693 of EGFR correlates uniquely with RTK recycling suggesting a functional cross talk between FGF and EGF signaling pathways. This finding is the key one that the authors focus on. They verify it in different systems, including an organoid one and use a variety of functional follow ups (siRNA, inhibitors) to establish a crucial role for this phospho site. The authors drill down on the mechanism in a number of ways, most importantly with and EGFR point mutant for threonine 693. They build a 'reciprocal priming' model that summarizes the interaction of FGFR and EGFR via threonine 693.

I find the study very elegant and done at a very high level of expertise in all the different experimental aspects that are involved. Not only did the authors perform cutting edge proteomics and phosphoproteomics but they also did so in many different settings and perturbations. Far from leaving it at the resource stage, the authors discovered a key functional site, which they generalize successfully into their 'reciprocal priming' model, which seems to be generalizable.

We thank the reviewer for his or her detailed assessment of our work and we appreciate the reviewers' positive comments on our manuscript. We are pleased to read that the reviewer is impressed by the identification and characterization of the reciprocal priming between FGFR and EGFR. We have addressed the reviewers' comments in the revised version of our manuscript.

Minor points:

- although common in the field, the authors could consider using less jargon, i.e. abbreviations that the reader has to remember but many won't.

We have changed the text following the reviewers' suggestion.

- given the central importance of this EGFR site (T693), is it evolutionary conserved and how far?

We thank the reviewer for this intriguing and thought-provoking question. We generated a best fit motif that encapsulates the T693 peptide region of EGFR (see figure below), and found that the region was conserved in birds, amphibians, fish and mammals - when searched against 1000 orthologous sequences extracted from BLAST. The complete evolutionary tree searched is attached as a figure (.txt) for the benefit of the reviewer. The highly conserved nature of this site further underpins the importance of its role in regulating EGFR activity. We have not included this data in the paper, but we have referenced it at page 10, where the text reads:

“Besides regulating EGFR trafficking (Heisermann et al., 1990), T693 phosphorylation is a highly conserved residue (data not shown) and involved in the response to stress or to the anti-tumor agent cisplatin via p38 activation (Winograd-Katz & Levitzki, 2006; Zwang & Yarden, 2006).”

Figure for reviewer: EGFR T693 consensus motif.

- Can the authors speculate on structural consequences of this phosphorylation and how it might explain the cross talk?

We thank the reviewer for the useful suggestion of looking at the structural consequences of T693 phosphorylation. There is nothing reported about the structural alterations arising from phosphorylation at T693 of EGFR. However, T693 is contained within the juxta membrane domain of EGFR, known to be unfolded and therefore more flexible, responsible for assisting kinase domain interaction. Given the juxta membrane unfolded nature, alteration to its 3D topology is not predictable for phosphorylation at T693. Phosphorylation in the juxta membrane domain may facilitate allosteric kinase activation as well as recruitment of downstream adaptors such PLCy1 (Thiel and Carpenter, PNAS, 2007). Although it is not possible to speculate on the structural changes arising from phosphorylation at T693 we can be confident of the regulatory importance of phosphorylation within the juxta membrane domain.

- 'Results' appears twice and there are no figure numbers.

We have corrected these mistakes and followed the guidelines for the preparation of figures.

Referee #2:

In this study the authors have used quantitative phosphoproteomics to analyse the signalling from the FGF receptor tyrosine kinase that is regulated by endocytic recycling. They have discovered that FGFR ligands that promote FGFR recycling induce phosphorylation of another receptor tyrosine kinase, the EGFR, on Threonine 693 and this phosphorylation modulates both EGFR and FGFR trafficking and signalling outputs. These findings are novel and intriguing and may have relevance to breast cancer, where targeting a single tyrosine kinase has limited efficacy. Understanding the interplay between different receptor kinases and how they are regulated by endocytic recycling may lead to the exploration of novel therapeutic avenues. Additionally, this study provides valuable data sets that will be of interest to multiple researchers in the fields of receptor tyrosine kinase signalling and trafficking.

We appreciate the positive comments of the reviewer on the novelty of our findings, including the potential relevance for breast cancer, and the validity of our datasets for the signalling and trafficking communities. We thank the reviewer for all useful comments and suggestions that we have addressed in the revised version of our manuscript.

Overall the study is well presented, the data convincing and represents an impressive mix of proteomics, microscopy and biochemical analysis. I am convinced that phosphorylation of EGFR on T693 requires FGFR recycling and that this phosphorylation regulates both FGFR and EGFR trafficking and signalling. Where I am less convinced is that the EGFR phosphorylation occurs in recycling endosomes. It is clear that FGF10-stimulated FGFR2 recycling is required for phosphorylation of EGFR T693 (FGF7, which doesn't promote FGFR2 recycling doesn't induce T693 phosphorylation, and T693 phosphorylation is inhibited by TTP or RCP depletion or expression of dom -ve Rab11, all of which inhibit FGF10-mediated FGFR2 recycling). Erk activation is necessary for T693 phosphorylation (as it is prevented by Erk inhibitors) but is not sufficient because FGF7 promotes Erk activation without promoting EGFR T693 phosphorylation. Notably, FGF10-mediated Erk activation appears to require recycling as it is inhibited by depletion of TTP and RCP. For me this latter result is extremely intriguing. Ideally, I would like to see as an additional control the effects of dominant negative dynamin expression on FGF10-mediated Erk activation. When recycling is inhibited with TTP or RCP there appears in some cases to be some degradation of FGFR which could contribute to inefficient Erk activation so it would be nice to check that inhibition of endocytosis (when there should be no loss of FGFR) also blocks FGF10-mediated Erk activation. If the authors have already done this it should be shown.

The authors suggest that EGFR T693 phosphorylation occurs in recycling endosomes, as evidenced by anti-phT693 staining in endosomes by IF in FGF10 stimulated cells. I think it is also possible that phosphorylation of EGFR on T693 occurs at the plasma membrane leading to EGFR endocytosis and accumulation in recycling endosomes. I think that the data is consistent with this possibility which is acknowledged in the discussion, but the summary states: "We discovered reciprocal priming between FGFRs and Epidermal Growth Factor Receptor (EGFR) within recycling endosomes".

Dom negative dynamin also blocks T693 phosphorylation - this would be expected to accumulate both FGFR and EGFR on the cell surface, supporting the idea that EGFR:PDGFR priming might occur in recycling endosomes but dominant negative dynamin could prevent the Erk activation that is also necessary for T693 phosphorylation.

We agree with the reviewer that T693 phosphorylation requires FGFR recycling and that it may occur when EGFR is present either in recycling endosomes or at plasma membrane. We have added the following data to support this idea and have changed the figures and the text accordingly. This new data still supports the two major findings of our study: that FGFR signaling from the recycling endosomes leads to phosphorylation of EGFR at T693, and that EGFR presence in the recycling endosomes alters the timing of FGFR trafficking.

The specific concerns raised by the reviewer have been addressed as follows.

1. The effects of dominant negative dynamin on ERK phosphorylation and on phosphorylation of EGFR on T693.

ERK phosphorylation does not change in breast cancer cells expressing dominant negative dynamin and stimulated with FGF10, TGF α or EGF. We have incorporated the data in Appendix Figure S4H and G.

The text in the results section now reads:

“Finally, FGF10-mediated phosphorylation of EGFR_T693 decreased when FGFR2b trafficking was inhibited by dominant negative Rab11 (preventing recycling) or dominant negative Dynamin (preventing internalization) *with no discernible alterations in ERK activation* (Appendix Figure S4E-H). As TGF α or EGF induced EGFR_T693 phosphorylation occurs regardless of length of stimulation, or trafficking inhibition (Figure 2J, Appendix Figure S4) *we concluded that EGFR_T693 phosphorylation is independent of EGFR recycling, but FGFR2b-induced phosphorylation of EGFR on T693 requires recycling upon FGF10 stimulation.*”

The figure legend now reads:

“G, H) Lysates from T47D and BT20 transfected with EGFP-Rab11, EGFP-Rab11_S25N (dominant negative Rab11), or Dynamin_K44A-GFP (dominant negative Dynamin) and stimulated or not with FGF10 or TGF α or EGF for either 8 or 40 min. as indicated, were immunoblotted with the indicated antibodies.”

All the considered FGFR and EGFR ligands were able to induce ERK phosphorylation. EGF and TGF α induced T693 phosphorylation regardless of the localization of EGFR and partially depending

on ERK activation (Figures 2, 3, and Appendix Figure S4). The FGFR2 ligand FGF7 did not induced T693 phosphorylation but induced ERK activation (Figures 2, 3, EV3). Finally, ERK phosphorylation was necessary but not sufficient to induce FGF10-dependent phosphorylation of T693 on EGFR. As FGFR2b is present in different endocytic vesicles upon FGF7 or FGF10 stimulation (Figure 1B) one possibility is that different pools of phosphorylated ERK exist depending on different sub-cellular compartments which drive the phosphorylation of specific residues in a stimulus-dependent manner. This idea is consistent with the lack of ERK and T693 phosphorylation in RCP- and TTP-depleted cells, where FGFR2b is partially degraded (see also below) (Figures 2 and Appendix Figure S4).

2. FGFR-mediated phosphorylation of EGFR on T693 happens in recycling endosomes:

Fractionation of cells into a membrane and a cytoplasmic fraction showed that FGF10 could induced phosphorylation of EGFR at T693 in a minor population of EGFR at plasma membrane, whereas the majority of EGFR phosphorylation on T693 occurred within the cytoplasmic fraction (Appendix Figure S4D below). Although technical restraints caused non pure fractions the phosphorylation of EGFR at T693 appears to be primarily of intracellular origin. This finding does not rule out membrane populations of phosphorylated EGFR accumulating in the cytoplasm. Nevertheless, it strongly implicates internal pools as the majority of observable FGF10-induced T693 phosphorylation of EGFR.

The text at page 6,7 now reads:

“Furthermore, we observed a peak in EGFR_T693 phosphorylation at 40 min. post-stimulation with FGF10 - when FGFR2b was present in recycling endosomes - and the majority of phosphorylated EGFR on T693 was found in the cytoplasm (Figure 1B-C-H-I, Appendix Figure S4C-D).”

The figure legend now reads:

“(D) Lysates of T47D treated with FGF10 for indicated time points following fractionation into membrane and cytoplasm immunoblotted with the indicated antibodies”

As EGFR was detected at plasma membrane in cells stimulated with FGF10 for 20 min. (Figure 5E) we checked the phosphorylation of T693 at this time point also by staining with an anti-phosphorylated T693 antibody followed by confocal analysis and quantification (new Figures 5D and Appendix Figure S6 below). We found that the majority of EGFR is not phosphorylated on T693 in cells stimulated for 20 min. with FGF10, when EGFR is detected at the plasma membrane. On the contrary, TGF α induced phosphorylation of EGFR at T693 also upon 20 min. stimulation with EGFR detected in the recycling compartment. Therefore, the majority of phosphorylation of EGR at T693 occurred between 20 and 40 min. stimulation with FGF10 when EGFR begins accumulating in FGFR2b-positive recycling endosomes.

The text at page 8 now reads:

“The FGFR/EGFR interplay was also verified by uncovering the co-localization of FGFR2b with EGFR in recycling endosomes upon 40 min. stimulation with FGF10, but not with TGFα (Figure 5A-B, Appendix Figure S6A-B). Intriguingly, EGFR was phosphorylated on T693 in recycling endosomes at 40, but not at 20 min., in response to FGF10 (Figure 5E-F, Appendix Figure S6C). We therefore hypothesized that the recycling endosomes may form the interface for FGFR and EGFR signal integration, perhaps involving physical interaction of the receptors, and that EGFR_T693 phosphorylation could affect FGFR2b trafficking. We assessed FGFR2b/EGFR co-localization and co-immunoprecipitation in T47D cells depleted of endogenous EGFR and transfected with either EGFR wild type (wt) or the T693A mutant (T693A) - which cannot be phosphorylated at residue T693 - upon a time-course stimulation with FGF10 (Figure 5, Appendix Figure S6, Appendix Figure S7A-B). Under non-stimulated conditions FGFR2b co-localized at the plasma membrane and co-immunoprecipitated with both wt and T693A EGFR (Figure 5E-G, Appendix Figure S6D-F). At 20 min. stimulation with FGF10, FGFR2b was detected in the recycling endosomes in both wt- and T693A-expressing cells, but it failed to interact or localize with either EGFR as both were still located at the plasma membrane (Figure 5E-F, Figures EV1-2, Appendix Figure S6G)”

To further narrow down the origin of the intracellular population of phosphorylation EGFR T693 we performed biotinylating pull down experiments using cells expressing GFP-Rab11-APEX2. The transfection of T47D with the GFP-Rab11-APEX2 construct did not alter the trafficking of FGFR2b, as

shown by the quantification in Appendix Figure S6G (see below), when compared to quantification of wild type T47D shown in Figure 1B. We found RAB25 and RCP, which are known binding partners of Rab11, in the “proximal to Rab11” samples but we did not find the nuclear marker Histone H3 (Appendix Figure S6H below). GFP-Rab11-APEX2 is within close proximity to the majority of FGF10-induced EGFR phosphorylated on T693, confirming the confocal results shown in Figure 5A, C, and E. Due to technical restraints of not having pure fractions of biotinylated samples some population of EGFR phosphorylated at T693 still exists either beyond the boundary of RAB11 recycling endosomes (possibly at the plasma membrane) or as contamination of incomplete exclusion of biotinylated proteins (see the results of blotting with anti-Strepavidin HRP).

Nevertheless, these results strongly implicate that the majority of FGF10-induced EGFR phosphorylated on T693 is found primarily in the proximity of the recycling endosomes.

The text at page 9 now reads:

“Using GFP-Rab11-APEX2, which did not affect FGFR2b trafficking (Appendix Figure S6H), we assessed at which time point the majority of EGFR phosphorylated on T693 was detected in proximity to the recycling endosomes following FGF10 stimulation. In agreement with the confocal imaging, we found that EGFR phosphorylation on T693 accumulated at the recycling endosomes between 20 and 40 min. stimulation, when FGFR2b was detected in recycling endosomes together with RCP

(Appendix Figure S6G-I, Figure 5). Altogether, these data suggest that EGFR_T693 phosphorylation dynamically regulates FGFR2b trafficking after the formation of an FGFR2b/EGFR/RCP complex in the recycling endosomes. Indeed, in T693A-expressing and in cells treated with FGFR and ERK inhibitors, but not EGFR or p38 inhibitors, there is less intracellular FGFR2b and increased receptor at the cell surface at 40 min. stimulation (Figure 5H, Appendix Figure S6J-K). Therefore, FGF10-dependent EGFR_T693 phosphorylation via FGFR and ERK activation (Figure 3) plays a role in the spatio-temporal regulation of FGFR2b trafficking.”

The figure legends now read:

“(I) T47D transfected with GFP-Rab11-APEX2 were stimulated with FGF10 at indicated timepoints. Biotinylation was performed followed by streptavidin bead pull-down, running the supernatant against the lysates extracted off the beads and immunoblotted with the indicated antibodies, including known Rab11 interactors RAB25 and RCP and nuclear control Histone H3.”

We have also added the following to the Material and Methods section:

“Cell fractionation and Biotinylation assays

Membrane and cytoplasmic fractionation was performed using Mem-PER™ Plus Membrane Protein Extraction Kit (ThermoFisher Scientific) according to the manufactures instructions. Biotinylation pull downs experiments were performed as described previously (Lobingier et al, 2017). Briefly biotinylation experiments were performed by transfecting GFP-Rab11-APEX2 constructs in to 2 million cells plated in 10cm dishes. Cells were pre-incubated (40 min.) with Biotin Phenol (Iris Biotech), after stimulation with ligands, Hydrogen peroxide (Sigma Aldrich) was added for 1 min. before quenching with Trolox (Sigma Aldrich) and Sodium ascorbate (VWR) during ice cold lysis. A 2 hour RT pull-down with streptavidin beads was then performed running the supernatant against the bound proteins.”

Combining the results of these experiments with the following:

1. Only ligands inducing recycling of their receptor lead to EGFR phosphorylation on T693 (Figure 2)
2. When FGFR and EGFR remains at the plasma membrane in dominant negative dynamin-expressing cells, FGF-induced EGFR phosphorylation on T693 was lost (Appendix Figure S4)

we concluded that FGF-induced EGFR phosphorylation on T693 required FGFR recycling and accumulation of EGFR in the recycling endosomes.

As highlighted by the reviewer we acknowledged the possibility of T693 phosphorylation occurring at plasma membrane as well as in recycling endosomes in the discussion which has been expanded and now reads:

“The dynamics of RTK trafficking is affected by several factors and in turns affects downstream signalling. For instance, EGFR recycles through recycling endosomes even in the absence of stimuli (Baumdick et al, 2015) and can be found in a subset of perinuclear compartments (Tomas et al, 2015). This potentially explains why the majority of EGFR_T693 phosphorylation is found at recycling endosomes upon FGF10 stimulation between 20 and 40 min., a time point when FGFR2b itself starts accumulating in the recycling endosomes. In turn, FGFR2b trafficking is deregulated when EGFR remains at the plasma membrane, e.g. in the absence of T693 phosphorylation, suggesting that the transient formation of the EGFR/FGFR2b complex on recycling endosomes is the key regulatory

event for the correct timing of FGR2b trafficking. One alternative model is that FGFR2b is internalized, traffics back to the plasma membrane where it retrieves EGFR, then reinternalizes (potentially recycling together EGFR), but the reinternalization step stalls without phosphorylation at T693. Another possibility is that FGFR2b phosphorylates the EGFR to prevent its recycling to the plasma membrane, thus explaining the change in EGFR distribution. This process should be replicated in FGF7-stimulated cells and would require further validation by high-resolution imaging of pools of FGFR2b and EGFR from the plasma membrane to recycling endosomes and back. The phosphorylation of EGFR might delay FGFR recycling back to the plasma membrane, thus allowing the formation of specific signaling complexes at the recycling endosomes. The kinases AKT and PAK1 identified by the three TPAs experiments would be interesting candidates to focus on to test this hypothesis. The alteration of the kinase landscape shown in T693A-expressing cells (Figure 6D) also confirms the idea that the EGFR phosphorylation may work as a scaffold for the recruitment of recycling machinery (e.g. RCP) and signaling partners to FGFR2b. “

To better reflect the conclusions of our study we have changed the summary as follow:

“Integration of signaling downstream of individual Receptor Tyrosine Kinases (RTKs) is crucial to fine tune cellular homeostasis during development and in pathological conditions, including breast cancer. However, how signaling integration is regulated and whether the endocytic fate of single receptors controls such signalling integration still remain poorly elucidated. Focusing on distinct Fibroblast Growth Factor Receptors (FGFRs) we generated a detailed picture of recycling-dependent FGF signaling in breast cancer cells by combining quantitative phosphoproteomics and targeted assays. We discovered reciprocal priming between FGFRs and Epidermal Growth Factor Receptor (EGFR) co-ordinated from recycling endosomes. FGFR recycling ligands induce EGFR phosphorylation on threonine 693. This phosphorylation event alters both FGFR and EGFR trafficking and primes FGFR-mediated proliferation but not cell invasion. In turn, FGFR signalling primes EGF-mediated outputs via threonine 693 phosphorylation. The discovery of reciprocal priming between distinct families of RTKs from recycling endosomes elucidates a novel signalling integration hub where recycling endosomes orchestrate cellular behaviour. Therefore, targeting reciprocal priming over individual receptors may improve personalized therapies in breast and other cancers.”

and altered the title to

“Reciprocal Priming between RTKs from Recycling Endosomes Orchestrates Cellular Signaling Outputs”.

I would very much like to be convinced that this priming event occurs in endosomes, but I am not sure that this has been conclusively shown. Nevertheless, I am convinced that reciprocal priming between EGFR and FGFR is occurring that depends on recycling. I think therefore that the authors should be very careful in their conclusions and, given that there is a lot of data in this paper, a summary table of the conditions under which FGFR recycles, Erk is activated, EGFR is phosphorylated on T693 etc, EGFR co-ips with FGFR might help. A summary figure is presented but I think that there are some assumptions made in that figure, eg has RCP been shown to mediate FGFR interaction with EGFR? Also a direct interaction between EGFR and FGFR on the plasma membrane is implied. Has this been shown (other than by co-ip which does not demonstrate a direct interaction)?

We are grateful for the suggestion of adding a summary table that we have included as EV3:

Observation	Conditions	Figures
FGF7 doesn't cause recycling of FGFR2b	confocal imaging, over expression of FGFR2b	Figure 1
FGF10 induces internalisation and recycling of FGFR2b	confocal imaging, membrane cytoplasm fractionation, Rab11-APEX2 biotinylation, co-IP	Figures 1 and 5, Appendix Figures S4 and S6
Phosphorylation of EGFR at T693 downstream of FGFRs is found in recycling conditions	three phosphoproteomics mass spec experiments (TPA1,2 and 3), Western Blots in several cell lines and organoid, over expression of FGFR2b.	Figures 2-5, Appendix Figures S2-5
Phosphorylation of EGFR at T693 relies on FGFR signalling through ERK	inhibitor experiments	Figures 3-4
Phosphorylation of EGFR at T693 downstream of FGFRs requires active recycling	RCP/TTP depletion experiments, dominant negative dynamin experiments	Figure 2, Appendix Figure S4
Pre-treatment of FGF10 leads to enhanced ERK activity downstream of EGFR stimulation	inhibitor experiments, western blots, qPCR of target genes, functional assays	Figure 4, Appendix Figure S5
FGF10 induced phosphorylation of T693 accumulates in the recycling endosomes after 20mins	three phosphoproteomics mass spec experiments (TPA1,2 and 3), confocal imaging, co-IPs, Rab11-APEX2 biotinylation experiment	Figures 2-5, Appendix Figures S4, S6-7
EGFR_T693A that can't internalise leads to faster FGFR2 trafficking back to PM	confocal imaging, co-IPs	Figure 5, Appendix Figure S6
EGFR_T693A that can't internalise alters FGFR2 signalling and cellular outputs	Phosphoproteomics, western blots, functional assays	Figures 5-7, Appendix Figures S6-7

The figure legend reads:

“Figure EV3. Table of key observations made in this study, a brief description of the conditions these observations were made under and the location of key results underpinning them.”

Furthermore, Figure 7G has been modified to better summarize the data described in Figure 5 (RCP/FGFR2b co-immunoprecipitation and FGFR2b/EGFR co-immunoprecipitation and co-localization at the plasma membrane). We have changed the figure legend from “known or discovered interaction” to “*known or discovered functional interaction*”. This suggests that EGFR and FGFR2b functionally cooperate at plasma membrane as they partially co-localize and they co-immunoprecipitate. We also indicated the possibility that FGFR2b may induce the phosphorylation of EGFR on T693 both at plasma membrane and in recycling endosomes (burgundy dotted arrow).

Additional points are detailed below:

Fig. 1H and I: following siRNA-mediated depletion of RCP or TTP, not only is FGF10-stimulated FGFR recycling inhibited but FGFR degradation appears to be promoted. This is potentially important as changes in phosphoproteome could be due to reduced numbers of FGFR in the cell, in addition to reduced accumulation in recycling endosomes. Is this reproducibly the case? If so it should be noted in the text.

We thank the reviewer for having pointed this out. We have discussed this point in the discussion which reads:

“The hypothesis of a multi-step regulation of FGFR2b - and possibly other RTKs - recycling is supported by our data on TTP and RCP. FGFR2b was not detected in recycling endosomes in TTP-depleted cells (Figure 1H), therefore we propose that TTP is required for FGFR2b entry into recycling endosomes in epithelial cells (Figure 1I) (Francavilla et al., 2013). This would explain the lack of T693 phosphorylation and FGFR2b degradation in the absence of TTP. The third player in the complex regulation of FGFR2b recycling is RCP, which is bound to FGFR2b and EGFR in the recycling endosomes. As FGFR2b localized to recycling endosomes in RCP-depleted cells (Figure 1H), but it is partially degraded in this condition (Figure 1I), RCP may play a role in FGFR2b exit from recycling endosomes. Therefore, RCP plays a temporally unique role in FGFR2b trafficking besides being a regulator of EGFR and integrin recycling (Caswell & Norman, 2008; Francavilla et al., 2016). We speculate on the presence of different populations of endosomes, either leading to recycling to plasma membrane (RCP-positive) or to receptor degradation (RCP-negative), implying that different families of RTKs regulate each other trafficking and signaling through a pool of adaptors on recycling endosomes. Future studies will reveal these adaptors upon different perturbations to build a more comprehensive regulatory network of RTK trafficking. “

I find the graphical presentation of the recycling assays (eg Fig 1I and Fig S3B) hard to read- the data points and error bars are difficult to make out.

We have changed the graphical representation in the following Figures and Appendix Figures: 1I, S1A, S1B, S3B, 5H, S6H, and S6I.

Fig. 3C The FGF10 treated cells expressing HA-FGFR2 appear to have far more FGFR2 than the other cells, which could be contributing to the elevated EGFR T693 phosphorylation.

We have substituted the WB of Figure 3C with a replicate from the same experiment. We apologized for the mistake.

The authors state that "EGFR_T693 phosphorylation is uniquely dependent on MEK-ERK signaling upon FGF10 but not TGF α stimulation." However, it appears from Fig. 3F that EGFR T693 phosphorylation is also substantially reduced following MEK inhibition in TGF α treated cells.

We have altered the text to better reflect the results of this experiment. Text at pages 7-8 now reads:

"FGF10 pre-treatment did not have the same effect on EGFR stabilization from the recycling stimulus TGF α at 120 min and a less pronounced effect on ERK phosphorylation at any time point. These data suggest that FGF10 pre-treatment alters EGF signaling to increase stability of EGFR at 120 min resulting in higher levels of ERK activation. FGFR2 levels remained high with all treatment conditions, however 40 min. FGF10 pre-treatment did alter the dynamics of FGFR2b downstream of both EGF and TGF α stimulation."

I have some concerns over the Western blot shown in Figure 4B which investigates the effects of pretreatment with FGF for 40 mins on EGFR abundance and signaling. I can see that FGF pretreatment increases the abundance of EGFR, but the effects on Erk phosphorylation appear somewhat marginal and the effect of FGF10 pretreatment appear even more clear on TGF stimulation than EGF stimulation, at variance with the authors assertion that "FGF10 pre-treatment did not affect the response of the EGFR recycling stimulus TGF". Also- FGFR levels appear to be majorly affected by treatment with EGF and TGF.

The reviewer is correct to point out that the pre-treatment with FGF10 enhances ERK phosphorylation downstream of both EGF and TGF α . Here, we wanted to highlight that EGFR levels were stabilised by FGF10 pre-treatment as opposed to degradation induced by EGF stimulation alone, whereas FGF10 pre-treatment followed by TGF α (inducing recycling of EGFR) did not substantially alter EGFR levels. The effects of pre-treatment on elevating ERK phosphorylation downstream of EGF were not always similar in the degree of amplitude, but they were convincingly reproducible (see Figures 4C-H and Appendix Figures S5A-C). We have altered the text to better reflect the full extent of these results of this experiment, and text now reads:

"FGF10 pre-treatment did not have the same effect on EGFR stabilization from the recycling stimulus TGF α at 120 min and a less pronounced effect on ERK phosphorylation at any time point. These data suggest that FGF10 pre-treatment alters EGF signaling to increase stability of EGFR at 120 min resulting in higher levels of ERK activation. FGFR2 levels remained high with all treatment conditions, however 40 min. FGF10 pre-treatment did alter the dynamics of FGFR2b downstream of both EGF and TGF α stimulation."

Additionally, in Figure 4E it appears to me that pretreatment with FGF does increase the level of EGFR following EGF stimulation in cells treated with EGFR inhibitor at variance with the authors statement that "Conversely, when EGFR was stabilized with AG1478 (Gan et al ,2007), the total level of EGFR was not affected by FGF10 pre-treatment although ERK activation decreased (Figure 4E). One of the potential issues here is that there are a lot of Western blots shown in this paper with no quantitation. I accept that Western blotting is only semi-quantitative at best, some of the Westerns are confirming data obtained from quantitative phosphoproteomics and some idea of reproducibility can be gleaned from the fact that there is some repetition of key treatments in different figures. Nevertheless, as detailed above, some of the Westerns, particularly in Figure 4, do not seem to agree with what is being described in the text.

We agree with the reviewer that Western blotting by itself is only semi-quantitative which is why most key points made in this manuscript are repeated in multiple figures in multiple different experiments, such as FGF10 stimulation leading to EGFR phosphorylation at T693 and pre-treatment with FGF10 leading to increase in ERK phosphorylation and EGFR levels. Indeed, we have also endeavoured to use other supporting methods where appropriate to strengthen the findings such as immunofluorescence, inhibitor experiments, rescue experiments and functional read outs. Specifically, Figure 4 shows not only western blots from multiple experiments as a read out of increased ERK activation following pre-treatment with FGF10, but also corroborating observations linked to downstream transcriptional events and cellular outputs. Where we have failed to adequately to describe the results, we have now altered the text to better reflect the full extent of our findings concerning figure 4:

"FGF10 Primes EGFR Responses

To explore the consequences of the FGFR and EGFR interplay we first tested the effect of FGF10 on EGFR functions. FGF10 stimulation did not alter the levels of EGFR, which decrease over time upon EGF - and to a lesser extent TGF α - stimulation (Francavilla et al., 2016) (Figure 4A). However, if T47D cells were pre-treated for 40 min. with FGF10, followed by stimulation with EGF for different time points, we observed an increase in EGFR abundance and ERK activation relative to cells not pretreated with FGF10 for 120 min. (Figure 4B). FGF10 pre-treatment did not have the same effect on EGFR stabilization from the recycling stimulus TGF α at 120 min and a less pronounced effect on ERK phosphorylation at any time point. These data suggest that FGF10 pre-treatment alters EGF signaling to increase stability of EGFR at 120 min resulting in higher levels of ERK activation. FGFR2 levels remained high with all treatment conditions, however 40 min. FGF10 pre-treatment did alter the dynamics of FGFR2b downstream of both EGF and TGF α stimulation. The effect of FGF10 pre-treatment on EGFR dynamics in EGF-stimulated cells depended on both FGFR and ERK activities (Figure 4C-D). Conversely, when EGFR was stabilized with AG1478 (Gan et al, 2007), the total level of EGFR was not affected by FGF10 pre-treatment although ERK activation decreased (Figure 4E). When EGFR was stabilized with AG1468 (Gan et al., 2007), the stability of the EGFR following pre-treatment was in line with FGF10 40 min. stimulation, whereas ERK activation decreased. Therefore, FGF10 pre-treatment increases the total levels of EGFR after EGF-stimulation, and this resulted in sustained ERK phosphorylation, an effect dependent on the activation of the FGFR-ERK signaling axis. Correspondingly, FGF10 pre-treatment followed by EGF stimulation for 4h induced the highest expression of ERK late target genes (Uhlitz et al, 2017) (Figure 4F), which may suggest enhanced cell cycle progression (Sharrocks, 2006). This was confirmed by increased EdU incorporation in T47D and BT20 stimulated with EGF upon pre-treatment with FGF10 (Figure 4G, Appendix Figure S5A). The use of specific inhibitors indicated that FGF10 significantly increased EGF-dependent cell cycle progression in an FGFR-, ERK-, and EGFR-dependent manner (Figure 4G-H and Appendix Figure S5).

In conclusion, FGF10 pre-treatment stabilizes EGF-stimulated EGFR via FGFR-ERK signaling. Elevated ERK phosphorylation, increased expression of ERK late target genes, and enhanced cell cycle progression suggest that FGF10 pre-treatment primes EGF responses and confirms a functional interplay between FGFR and EGFR signaling."

The authors state: "Indeed, FGFR2b trafficking is faster in T693A-expressing cells and in cells treated with FGFR and ERK inhibitors, but not EGFR or p38, inhibitors (Figure 5H, Appendix Figure S6H-I)." These figures show that there is reduced intracellular FGFR and increased cell surface receptor at 40 minutes but that this represents faster trafficking is an assumption. For example less FGFR could be endocytosed. Nevertheless, the possibility that phosphorylation of the EGFR delays recycling of FGFR, modifying signalling output is an interesting one. Can the authors speculate on how phosphorylation of EGFR on T693 can affect trafficking of FGFR?

We agree with the reviewer and changed the result section as: "*Indeed, in T693A-expressing and in cells treated with FGFR and ERK inhibitors, but not EGFR or p38 inhibitors, there is less intracellular FGFR2b and increased receptor at the cell surface at 40 min. stimulation (Figure 5H, Appendix Figure S6H-I).*"

We have also speculated about this point in the discussion:

"The phosphorylation of EGFR might delay FGFR recycling back to the plasma membrane, thus allowing the formation of specific signalling complexes at the recycling endosomes. The kinases AKT and PAK1 identified by the three TPAs experiments would be interesting candidates to focus on. The alteration of the kinase landscape shown in T693A-expressing cells (Figure 6D) also confirms the idea that the EGFR phosphorylation may work as a scaffold for the recruitment of recycling machinery (e.g. RCP) and signalling partners to FGFR2b."

Referee #3:

This is a detailed and creative look into the crosstalk between two receptor tyrosine kinases (RTK). The overarching conclusion of the manuscript is that stimulation of the FGFR with a ligand that induces degradation (FGF10) of the ligand:receptor complex primes the EGFR for enhanced signaling. This study is comprehensive, the data are generally of high quality, and the manuscript is well written. This is a high quality cell signaling study that will be appreciated by the readership of EMBO J.

The manuscript starts with a proteomics approach to look for changes in the phosphorylation profiles of cellular proteins following treatment with ligands that are known to promote receptor recycling (FGF7) versus receptor degradation (FGF10). From those studies, they identify the EGFR as a phospho-substrate following treatment with FGF10. Subsequent biochemical analysis reveals that the EGFR is phosphorylated on Threonine 693 in response to stimulation with FGF10 (degradation ligand). A series of knock down and rescue experiments argues for the need for the EGFR to be phosphorylated on Thr693. Detailed biochemical/pharmacological characterization demonstrates that this phosphorylation is mediated through FGFR2b and utilizes the MEK pathway. Further, phosphorylation of the EGFR on Thr693 slows the rate of ligand stimulated EGFR degradation and sustains its signaling. In addition, the authors include studies that demonstrate the FGFR2 and EGFR co-localize in early endosomes. In the absence of Thr693 phosphorylation (by expressing a phosphorylation defective mutant receptor: EGFR_T693A), there is reduced EGFR in endosomes. However, the ligand-stimulated FGFR2 and the EGFR can still associate. The authors propose a model in which EGFR phosphorylation at Thr693 dynamically regulates FGFR2b trafficking. Finally, the authors demonstrate that pretreatment with FGF10 can enhance DNA synthesis and cell invasion in response to an EGFR ligand (TGF α). Both invasion and DNA synthesis require EGFR phosphorylation at Thr693 for this enhanced response.

The major strengths of the manuscript are the multifaceted approaches to testing the hypothesis as well as the use of multiple cell lines. There are only two concerns regarding the data and its interpretation (described below). The biochemical findings are strengthened by the observed changes in cell biology (DNA synthesis, invasion).

We thank the reviewer for his/her comments on our study. We would kindly like to highlight that I. FGF10 induces the recycling and not the degradation of FGFR2b, as also shown by Belleudi et al., 2007 and Francavilla et al., 2013 (Figures 1B, 1H, S1A, S1B, 5A, 5B, 5E, EV1, EV2, S6); II. FGFR2b and EGFR co-localize on recycling not early endosomes (Figure 5); III. FGF10 induced DNA synthesis but not cell invasion depending on the phosphorylation of T693 (Figure 7). We answered below to the concerns raised by the reviewer.

There is a concern regarding the micrographs in Figure 5. The colocalization of the receptors are not clear, even with the magnifications that are presented. One suggestion is to show individual channels and the merge of this magnified section. This addition is unlikely to change the interpretation given the quantifications of the co-localization of the fluorophores that accompany the image, but will help the reader visualize the data better.

We have shown individual channels and the merge of each magnified section in Figure EV1 and EV2 with the following Figure Legends:

“Figure EV1: Individual panels and merge of each magnified section of T47D depleted of EGFR by siRNA followed by transfection with wt EGFR and stimulated or not with either FGF10 or TGF α for the indicated time periods. Scale bars are indicated in the figure. Panels from figure 5E are shown on the left for comparison.”

“Figure EV2: Individual panels and merge of each magnified section of T47D depleted of EGFR by siRNA followed by transfection with EGFR_T693A and stimulated or not with either FGF10 or TGF α for the indicated time periods. Scale bars are indicated in the figure. Panels from figure 5E are shown on the left for comparison.”

The results now read:

“At 20 min. stimulation with FGF10, FGFR2b was detected in the RE in both wt- and T693A-expressing cells, but it failed to interact or localize with either EGFR as both were still located at the PM (Figure 5E-F, Figure EV1-2, Appendix Figure S6G).”

Second, the kinetics of EGFR phosphorylation and internalization are unclear particularly when the constitutive recycling of the EGFR is considered. An alternative model for these data would be that the FGFR2b phosphorylates the EGFR to prevent its recycling to the plasma membrane, rather than promoting its internalization. Thus, the change in EGFR distribution is not due to an increase in receptor internalization, but from a slowed return to the plasma membrane. The kinetics would be more consistent with this model, but there may be additional evidence that refutes this model. If appropriate, this idea should be incorporated into the discussion.

We thank the reviewer for suggesting alternative models to interpret the kinetics of EGFR phosphorylation and internalization. We expanded this concept in the Discussion which now reads:

“The dynamics of RTK trafficking is affected by several factors and in turns affects downstream signaling. For instance, EGFR recycles through recycling endosomes even in the absence of stimuli (Baumdick et al, 2015) and can be found in a subset of perinuclear compartments (Tomas et al, 2015). This potentially explains why the majority of EGFR_T693 phosphorylation is found at recycling endosomes upon FGF10 stimulation between 20 and 40 min., a time point when FGFR2b itself starts accumulating in the recycling endosomes. In turn, FGFR2b trafficking is deregulated when EGFR remains at the plasma membrane, e.g. in the absence of T693 phosphorylation, suggesting that the transient formation of the EGFR/FGFR2b complex on recycling endosomes is the key regulatory event for the correct timing of FGR2b trafficking. One alternative model is that FGFR2b is internalized, traffics back to the plasma membrane where it retrieves EGFR, then reinternalizes (potentially recycling together EGFR), but the reinternalization step stalls without phosphorylation at T693. Another possibility is that FGFR2b phosphorylates the EGFR to prevent its recycling to the plasma membrane, thus explaining the change in EGFR distribution. This process should be replicated in FGF7-stimulated cells and would require further validation by high-resolution imaging of pools of FGFR2b and EGFR from the plasma membrane to recycling endosomes and back. The phosphorylation of EGFR might delay FGFR recycling back to the plasma membrane, thus allowing the formation of specific signaling complexes at the recycling endosomes. The kinases AKT and PAK1 identified by the three TPAs experiments would be interesting candidates to focus on to test this hypothesis. The alteration of the kinase landscape shown in T693A-expressing cells (Figure 6D) also confirms the idea that the EGFR phosphorylation may work as a scaffold for the recruitment of recycling machinery (e.g. RCP) and signaling partners to FGFR2b. The hypothesis of a multi-step regulation of FGFR2b - and possibly other RTKs - recycling is supported by our data on TTP and RCP. FGFR2b was not detected in recycling endosomes in TTP-depleted cells (Figure 1H), therefore we propose that TTP is required for FGFR2b entry into recycling endosomes in epithelial cells (Figure 1I) (Francavilla et al., 2013). This would explain the lack of T693 phosphorylation and FGFR2b degradation in the absence of TTP. The third player in the complex regulation of FGFR2b recycling is RCP, which is bound to FGFR2b and EGFR in the recycling endosomes. As FGFR2b localized to recycling endosomes in RCP-depleted cells (Figure 1H), but it is partially degraded in this condition (Figure 1I), RCP may play a role in FGFR2b exit from recycling endosomes. Therefore, RCP plays a temporally unique role in FGFR2b trafficking besides being a regulator of EGFR and integrin recycling (Caswell & Norman, 2008; Francavilla et al., 2016). We speculate on the presence of different populations of endosomes, either leading to recycling to plasma membrane (RCP-positive) or to receptor degradation (RCP-negative), implying that different families of RTKs regulate each other trafficking and signaling through a pool of adaptors on recycling endosomes. Future studies will reveal

these adaptors upon different perturbations to build a more comprehensive regulatory network of RTK trafficking.”

Furthermore, as highlighted in our response to reviewer 2, we set out the following:

FGFR-mediated phosphorylation of EGFR on T693 happens in recycling endosomes:

Fractionation of cells into a membrane and a cytoplasmic fraction showed that FGF10 could induced phosphorylation of EGFR at T693 in a minor population of EGFR at plasma membrane, whereas the majority of EGFR phosphorylation on T693 occurred within the cytoplasmic fraction (Appendix Figure S4D below). Although technical restraints caused non pure fractions the phosphorylation of EGFR at T693 appears to be primarily of intracellular origin. This finding does not rule out membrane populations of phosphorylated EGFR accumulating in the cytoplasm. Nevertheless, it strongly implicates internal pools as the majority of observable FGF10-induced T693 phosphorylation of EGFR.

The text at page 6,7 now reads:

“Furthermore, we observed a peak in EGFR_T693 phosphorylation at 40 min. post-stimulation with FGF10 - when FGFR2b was present in recycling endosomes - and the majority of phosphorylated EGFR on T693 was found in the cytoplasm (Figure 1B-C-H-I, Appendix Figure S4C-D).”

The figure legend now reads:

“(D) Lysates of T47D treated with FGF10 for indicated time points following fractionation into membrane and cytoplasm immunoblotted with the indicated antibodies”

As EGFR was detected at plasma membrane in cells stimulated with FGF10 for 20 min. (Figure 5E) we checked the phosphorylation of T693 at this time point also by staining with an anti-phosphorylated T693 antibody followed by confocal analysis and quantification (new Figures 5D and Appendix Figure S6 below). We found that the majority of EGFR is not phosphorylated on T693 in cells stimulated for 20 min. with FGF10, when EGFR is detected at the plasma membrane. On the contrary, TGFalpha induced phosphorylation of EGFR at T693 also upon 20 min. stimulation with EGFR detected in the recycling compartment. Therefore, the majority of phosphorylation of EGR at T693 occurred between 20 and 40 min. stimulation with FGF10 when EGFR begins accumulating in FGFR2b-positive recycling endosomes.

The text at page 8 now reads:

“The FGFR/EGFR interplay was also verified by uncovering the co-localization of FGFR2b with EGFR in recycling endosomes upon 40 min. stimulation with FGF10, but not with TGFα (Figure 5A-B, Appendix Figure S6A-B). Intriguingly, EGFR was phosphorylated on T693 in recycling endosomes at 40, but not at 20 min., in response to FGF10 (Figure 5E-F, Appendix Figure S6C). We therefore hypothesized that the recycling endosomes may form the interface for FGFR and EGFR signal integration, perhaps involving physical interaction of the receptors, and that EGFR_T693 phosphorylation could affect FGFR2b trafficking. We assessed FGFR2b/EGFR co-localization and co-immunoprecipitation in T47D cells depleted of endogenous EGFR and transfected with either EGFR wild type (wt) or the T693A mutant (T693A) - which cannot be phosphorylated at residue T693 - upon a time-course stimulation with FGF10 (Figure 5, Appendix Figure S6, Appendix Figure S7A-B). Under non-stimulated conditions FGFR2b co-localized at the plasma membrane and co-immunoprecipitated with both wt and T693A EGFR (Figure 5E-G, Appendix Figure S6D-F). At 20 min. stimulation with FGF10, FGFR2b was detected in the recycling endosomes in both wt- and T693A-expressing cells, but it failed to interact or localize with either EGFR as both were still located at the plasma membrane (Figure 5E-F, Figures EV1-2, Appendix Figure S6G)”

To further narrow down the origin of the intracellular population of phosphorylation EGFR T693 we performed biotinylating pull down experiments using cells expressing GFP-Rab11-APEX2. The transfection of T47D with the GFP-Rab11-APEX2 construct did not alter the trafficking of FGFR2b, as shown by the quantification in Appendix Figure S6G (see below), when compared to quantification of

wild type T47D shown in Figure 1B. We found RAB25 and RCP, which are known binding partners of Rab11, in the “proximal to Rab11” samples but we did not find the nuclear marker Histone H3 (Appendix Figure S6H below). GFP-Rab11-APEX2 is within close proximity to the majority of FGF10-induced EGFR phosphorylated on T693, confirming the confocal results shown in Figure 5A, C, and E. Due to technical restraints of not having pure fractions of biotinylated samples some population of EGFR phosphorylated at T693 still exists either beyond the boundary of RAB11 recycling endosomes (possibly at the plasma membrane) or as contamination of incomplete exclusion of biotinylated proteins (see the results of blotting with anti-Streptavidin HRP).

Nevertheless, these results strongly implicate that the majority of FGF10-induced EGFR phosphorylated on T693 is found primarily in the proximity of the recycling endosomes.

The text at page 9 now reads:

“Using GFP-Rab11-APEX2, which did not affect FGFR2b trafficking (Appendix Figure S6H), we assessed at which time point the majority of EGFR phosphorylated on T693 was detected in proximity to the recycling endosomes following FGF10 stimulation. In agreement with the confocal imaging, we found that EGFR phosphorylation on T693 accumulated at the recycling endosomes between 20 and 40 min. stimulation, when FGFR2b was detected in recycling endosomes together with RCP

(Appendix Figure S6G-I, Figure 5). Altogether, these data suggest that EGFR_T693 phosphorylation dynamically regulates FGFR2b trafficking after the formation of an FGFR2b/EGFR/RCP complex in the recycling endosomes. Indeed, in T693A-expressing and in cells treated with FGFR and ERK inhibitors, but not EGFR or p38 inhibitors, there is less intracellular FGFR2b and increased receptor at the cell surface at 40 min. stimulation (Figure 5H, Appendix Figure S6J-K). Therefore, FGF10-dependent EGFR_T693 phosphorylation via FGFR and ERK activation (Figure 3) plays a role in the spatio-temporal regulation of FGFR2b trafficking.”

The figure legends now read:

“(I) T47D transfected with GFP-Rab11-APEX2 were stimulated with FGF10 at indicated timepoints. Biotinylation was performed followed by streptavidin bead pull-down, running the supernatant against the lysates extracted off the beads and immunoblotted with the indicated antibodies, including known Rab11 interactors RAB25 and RCP and nuclear control Histone H3.”

We have also added the following to the Material and Methods section:

“Cell fractionation and Biotinylation assays

Membrane and cytoplasmic fractionation was performed using Mem-PER™ Plus Membrane Protein Extraction Kit (ThermoFisher Scientific) according to the manufactures instructions. Biotinylation pull downs experiments were performed as described previously (Lobingier et al, 2017). Briefly biotinylation experiments were performed by transfecting GFP-Rab11-APEX2 constructs in to 2 million cells plated in 10cm dishes. Cells were pre-incubated (40 min.) with Biotin Phenol (Iris Biotech), after stimulation with ligands, Hydrogen peroxide (Sigma Aldrich) was added for 1 min. before quenching with Trolox (Sigma Aldrich) and Sodium ascorbate (VWR) during ice cold lysis. A 2 hour RT pull-down with streptavidin beads was then performed running the supernatant against the bound proteins.”

Combining the results of these experiments with the following:

1. Only ligands inducing recycling of their receptor lead to EGFR phosphorylation on T693 (Figure 2)
2. When FGFR and EGFR remains at the plasma membrane in dominant negative dynamin-expressing cells, FGF-induced EGFR phosphorylation on T693 was lost (Appendix Figure S4)

we concluded that FGF-induced EGFR phosphorylation on T693 required FGFR recycling and accumulation of EGFR in the recycling endosomes.

As highlighted by the reviewer we acknowledged the possibility of T693 phosphorylation occurring at plasma membrane as well as in recycling endosomes in the discussion which has been expanded and now reads:

“The dynamics of RTK trafficking is affected by several factors and in turns affects downstream signalling. For instance, EGFR recycles through recycling endosomes even in the absence of stimuli (Baumdick et al, 2015) and can be found in a subset of perinuclear compartments (Tomas et al, 2015). This potentially explains why the majority of EGFR_T693 phosphorylation is found at recycling endosomes upon FGF10 stimulation between 20 and 40 min., a time point when FGFR2b itself starts accumulating in the recycling endosomes. In turn, FGFR2b trafficking is deregulated when EGFR remains at the plasma membrane, e.g. in the absence of T693 phosphorylation, suggesting that the transient formation of the EGFR/FGFR2b complex on recycling endosomes is the key regulatory event for the correct timing of FGR2b trafficking. One alternative model is that FGFR2b is internalized,

traffics back to the plasma membrane where it retrieves EGFR, then reinternalizes (potentially recycling together EGFR), but the reinternalization step stalls without phosphorylation at T693. Another possibility is that FGFR2b phosphorylates the EGFR to prevent its recycling to the plasma membrane, thus explaining the change in EGFR distribution. This process should be replicated in FGF7-stimulated cells and would require further validation by high-resolution imaging of pools of FGFR2b and EGFR from the plasma membrane to recycling endosomes and back. The phosphorylation of EGFR might delay FGFR recycling back to the plasma membrane, thus allowing the formation of specific signaling complexes at the recycling endosomes. The kinases AKT and PAK1 identified by the three TPAs experiments would be interesting candidates to focus on to test this hypothesis. The alteration of the kinase landscape shown in T693A-expressing cells (Figure 6D) also confirms the idea that the EGFR phosphorylation may work as a scaffold for the recruitment of recycling machinery (e.g. RCP) and signaling partners to FGFR2b. “

To better reflect the conclusions of our study we have changed the summary as follow:

“Integration of signaling downstream of individual Receptor Tyrosine Kinases (RTKs) is crucial to fine tune cellular homeostasis during development and in pathological conditions, including breast cancer. However, how signaling integration is regulated and whether the endocytic fate of single receptors controls such signalling integration still remain poorly elucidated. Focusing on distinct Fibroblast Growth Factor Receptors (FGFRs) we generated a detailed picture of recycling-dependent FGF signaling in breast cancer cells by combining quantitative phosphoproteomics and targeted assays. We discovered reciprocal priming between FGFRs and Epidermal Growth Factor Receptor (EGFR) co-ordinated from recycling endosomes. FGFR recycling ligands induce EGFR phosphorylation on threonine 693. This phosphorylation event alters both FGFR and EGFR trafficking and primes FGFR-mediated proliferation but not cell invasion. In turn, FGFR signalling primes EGF-mediated outputs via threonine 693 phosphorylation. The discovery of reciprocal priming between distinct families of RTKs from recycling endosomes elucidates a novel signalling integration hub where recycling endosomes orchestrate cellular behaviour. Therefore, targeting reciprocal priming over individual receptors may improve personalized therapies in breast and other cancers.”

and altered the title to

“Reciprocal Priming between RTKs from Recycling Endosomes Orchestrates Cellular Signaling Outputs”.

Minor concerns

In Figure 1H the nuclei are quite variable in size. For instance, the cells treated with FGF10 for 40 minutes have nuclei that are 3-4 times the size of the cells treated with FGF10 for 120 minutes. Is there a scientific reason or technical reason? Are there more representative images?

We have substituted the two mentioned panels with more representative pictures in Figure 1H to make these panels consistent with those shown in Figure 1B at the same time points.

The figure legends for Fig 5H do not explain how the quantifications were obtained. Are these derived from the micrographs? How many cells and experiments do they represent.

The figure legend now reads: “The presence (total), internalization (internalized), and recycling (cell surface) of FGFR2 in T47D depleted of EGFR by siRNA followed by transfection with wt or T693A

and stimulated with FGF10 for different time periods were quantified as described (Francavilla *et al.*, 2016) and in the section 'Quantification of the Recycling Assay'. Briefly, we assessed approximately 100 cells per condition and expressed the results as the percentage of receptor-positive cells over total cells (corresponding to DAPI-stained nuclei) and referred to the values obtained at time zero. Values represent the median \pm SD of N=3."

Thank you for submitting your revised manuscript to The EMBO Journal. It has now been re-reviewed by referees 2 and 3, who found the original concerns satisfactorily addressed and have now further reservations regarding publication. Following clarification of minor remaining referee points and incorporation of the editorial issues listed below, we shall therefore be happy to accept the study for The EMBO Journal!

Referee #2:

The authors have done a great job in the revised manuscript and satisfied all my queries. I just seek clarification of one point. I am confused by the fractionation shown in Fig. S4D. Does the Mem-PER[®] Plus Membrane Protein Extraction Kit (ThermoFisher Scientific) distinguish between surface and intracellular membranes? I looked up the kit but it seemed a bit ambiguous, and I couldn't see how it was distinguishing between the two types of membrane. This is a relatively minor point because the immunofluorescence data and the new biotinylation data make the point very nicely that the majority of EGFR phosphorylated on Thr693 is in recycling endosomes. Therefore, unless the authors can provide suitable evidence or clarification on this point, then I think it is not essential for the paper.

Referee #3:

This is a detailed and creative look into the crosstalk between two receptor tyrosine kinases (RTK) specifically FGFR2b and EGFR. The manuscript builds on data from the proteomic analysis and identifies EGFR phosphorylation at Threonine 693 as an event that occurs specifically in response to signaling by recycling receptors. The authors examine spatial and temporal regulation of phosphorylation and provide a detailed biochemical analysis of the basis of the EGFR phosphorylation. These studies are complemented with assays that delineate the functional consequence of EGFR phosphorylation. This is an important study that will serve as a template for future studies exploring the crosstalk between signaling receptors, as these findings are likely not unique to the FGFR2b and the EGFR.

The authors provide a comprehensive response to the original critiques. There are no concerns about the resubmitted manuscript.

Point-by-point responses to reviewer's comments.

Our responses to reviewer comments are provided below in blue font after each comment.

Referee #2:

The authors have done a great job in the revised manuscript and satisfied all my queries. I just seek clarification of one point. I am confused by the fractionation shown in Fig. S4D. Does the Mem-PER[®] Plus Membrane Protein Extraction Kit (ThermoFisher Scientific) distinguish between surface and intracellular membranes? I looked up the kit but it seemed a bit ambiguous, and I couldn't see how it was distinguishing between the two types of membrane. This is a relatively minor point because the immunofluorescence data and the new biotinylation data make the point very nicely that the majority of EGFR phosphorylated on Thr693 is in recycling endosomes. Therefore, unless the authors can provide suitable evidence or clarification on this point, then I think it is not essential for the paper.

We thank the reviewer for the positive assessment for our revised manuscript. As pointed out by the reviewer the kit does not allow distinguishing between surface and intracellular membranes. Therefore, following the suggestion of the reviewer, we have removed Figure S4D and renamed the panels in the text and in the SI file. We have also removed the description of the experiment from the Material and Methods.

Referee #3:

This is a detailed and creative look into the crosstalk between two receptor tyrosine kinases (RTK) specifically FGFR2b and EGFR. The manuscript builds on data from the proteomic analysis and identifies EGFR phosphorylation at Threonine 693 as an event that occurs specifically in response to signaling by recycling receptors. The authors examine spatial and temporal regulation of phosphorylation and provide a detailed biochemical analysis of the basis of the EGFR phosphorylation. These studies are complemented with assays that delineate the functional consequence of EGFR phosphorylation. This is an important study that will serve as a template for future studies exploring the crosstalk between signaling receptors, as these findings are likely not unique to the FGFR2b and the EGFR.

The authors provide a comprehensive response to the original critiques. There are no concerns about the resubmitted manuscript.

We thank the reviewer for the positive assessment for our revised manuscript.

Corresponding Author Name: Chiara Francavilla

Journal Submitted to: The EMBO Journal

Manuscript Number: EMBOJ-2020-107182